# Chromatin conformation regulates the coordination between DNA replication and transcription

Ricardo Almeida[1], José Miguel Fernández-Justel[1], Cristina Santa-María[1], Jean-Charles Cadoret[2], Laura Cano-Aroca[1], Rodrigo Lombraña[1], Gonzalo Herranz[1], Alessandra Agresti [3] & María Gómez [1]

Chromatin is the template for the basic processes of replication and transcription, making the maintenance of chromosomal integrity critical for cell viability. To elucidate how dividing cells respond to alterations in chromatin structure, here we analyse the replication programme of primary cells with altered chromatin configuration caused by the genetic ablation of the HMGB1 gene, or three histone H1 genes. We find that loss of chromatin compaction in H1-depleted cells triggers the accumulation of stalled forks and DNA damage as a consequence of transcription–replication conflicts. In contrast, reductions in nucleosome occupancy due to the lack of HMGB1 cause faster fork progression without impacting the initiation landscape or fork stability. Thus, perturbations in chromatin integrity elicit a range of responses in the dynamics of DNA replication and transcription, with different consequences on replicative stress. These findings have broad implications for our understanding of how defects in chromatin structure contribute to genomic instability.

[1] Functional Organization of the Genome Group, Centro de Biología Molecular Severo Ochoa CBMSO (CSIC/UAM), Nicolás Cabrera 1, 28049 Madrid, Spain. [2] Institut Jacques Monod, CNRS-Université Paris Diderot, 75205 Paris, France. [3] San Raffaele Scientific Institute, Via Olgettina 58, 20132 Milano, Italy. These authors contributed equally: Ricardo Almeida, José Miguel Fernández-Justel, Cristina Santa-María.  Correspondence and requests for materials should be addressed to M.Góm. (email: mgomez@cbm.csic.es)

Every time a cell divides, its entire genetic and epigenetic information must be accurately replicated. In eukaryotic cells this occurs during the S-phase of the cell cycle through the activity of hundreds to thousands of replication origins (ORIs) distributed along their large genomes, in a context of a tightly packaged chromatin structure. Chromatin encodes epigenetic information and governs genome stability by protecting DNA to mutagenic agents and by regulating the accessibility of protein complexes to DNA. Two outstanding recent reports have successfully reconstituted efficient and regulated budding yeast chromatin replication in vitro, providing important clues on the regulatory role of chromatin both in ORI specification and replisome progression[1,2].

The basic unit of chromatin is the nucleosome, in which 147 bp of duplex DNA are wrapped around a histone octamer containing two copies of each of the four core histones: H2A, H2B, H3 and H4[3]. The higher-order organization of nucleosome core particles is controlled by the association of the intervening DNA with either the linker histone H1 or with non-histone proteins such as the high mobility group box (HMGB) family, that seem to exert different effects (reviewed in refs. [4,5]). While histone H1 is believed to stabilize the nucleosome by preventing DNA unwrapping, HMGB proteins impose a bending of the DNA that might destabilize the nucleosome structure, facilitating its remodelling[6–9]. In agreement with this, histone H1 depletion alters global chromatin compaction in mammalian cells and causes de-repression of heterochromatin transposable elements in *Drosophila*[10–13]. On the other hand, HMGB1 depletion associates with reduced nucleosome occupancies and increased amounts of RNA transcripts both in *Saccharomyces cerevisiae* and in mammalian cells[14]. Interestingly, replisome progression studies with purified proteins on a chromatin template have found that Nhp6a, the yeast ortholog of mammalian HMGB1, additively stimulate the rate of replication in the presence of the histone chaperone FACT[2]. Here, by employing genetic ablation of three of the somatic isoforms of histone H1 or for HMGB1, we address how alterations in chromatin structure affect the definition of the sites of replication initiation and the kinetics of replication elongation in vivo. We find that histone H1 depletion generates massive replication fork stalling and DNA damage signalling as a consequence of transcription–replication conflicts, while the increased chromatin dynamics associated with HMGB1 depletion allows faster fork progression without altering the replication initiation landscape or generating fork instability. These findings illustrate that alterations in the DNA-histone ratio impact both on replication and transcription dynamics, unveiling how defects in chromatin structure lead to replication stress that could, in turn, enhance cellular aging and disease states during development.

## Results

### Reduced nucleosome numbers allow faster replication rates.
HMGB1-KO mice die soon after birth[15]. However, HMGB1-KO MEFs are viable (Supplementary Fig. 1a), and contain a reduced amount of all canonical histones and the variant H2A.X in chromatin, likely due to the missing nucleosome-assembly activity of HMGB1[14,16]. The decrease in nucleosome number does not alter nucleosome spacing or positioning, but results in a non-uniform reduction of nucleosome occupancy across the genome[14]. This means that nucleosomes spend less time in each position to maintain the coverage of all locations genomewide. Interestingly, the increased chromatin dynamics associate with an overall increase in transcript abundance and specific alterations in the expression of a subset of genes[14]. To investigate the effects of reduced nucleosome occupancies in genomic DNA replication,

we first analysed the replication initiation profile of HMGB1-KO cells by short nascent strand sequencing (SNS-Seq). This technique consists in the specific isolation and sequencing of short leading strands from asynchronously growing cells, generating a landscape of replication initiation probabilities within a cell population (Supplementary Fig. 2a) (reviewed in ref. [17]). SNS-Seq has been applied to identify ORIs in several systems[18–23], contributing to build up the notion that the usage of sites of replication initiation in large genomes is intrinsically flexible and related to the transcriptional and epigenetic programme. Analysis of SNS-Seq datasets derived from HMGB1-KO MEFs showed limited changes in the replication initiation profile of cells containing reduced number of nucleosomes relative to their wild-type (WT) counterparts (Figs. 1a and 2a). Note that the lower overlap between WT and HMGB1-KO ORIs (63%) relative to that obtained between replicates (78–80%) (Fig. 1b), might account for the differences in read coverage and consequently in ORI numbers between individual SNS-Seq experiments (Supplementary Fig. 2b). In spite of this, both ORI density (Fig. 1c), and ORI distribution across annotated genomic features (Fig. 1d), are highly correlated between both genotypes, recapitulating the ORI enrichment at CpG islands, transcription start sites (TSS) and exons reported previously in mouse cells[20,24–26].

These findings are in agreement with results from single-molecule analysis of replication intermediates (Fig. 1e), where no significant differences were detected in the inter-origin distances between WT and HMGB1-KO cells, implying no activation of additional ORIs in cells with reduced nucleosome content (Fig. 1f and Supplementary Fig. 1b and e). Strikingly, fork rate measurements detected a strong increase in replication fork speed in mutant cells (Fig. 1g and Supplementary Fig. 1c and e). This increase in the replication elongation rates, however, is not accompanied by an increment in fork instability, as concluded from detailed analysis of fork asymmetry (Fig. 1h and Supplementary Fig. 1d-e), and S-phase profiling (Supplementary Fig. 3a-c). The data indicates that in this context of a more dynamic chromatin, the replication initiation landscape is globally maintained and replication forks move faster with stable elongation dynamics.

### H1-depletion alters replication initiation and fork dynamics.
Mouse embryos lacking the histone genes H1c, H1d and H1e subtypes (H1-TKO, Supplementary Fig. 1f) contain a 50% reduction in the amount of histone H1, and die by E11.5 with a very broad range of defects[27]. ES cell lines derived from these embryos have globally a less compact chromatin structure but, unexpectedly, only the expression of a small number of genes is affected[10,28,29]. H1-TKO mES cells have the equivalent to a molar ratio of one H1 molecule per four nucleosome cores and a 15 bp reduction in the spacing between nucleosomes[10]. In striking contrast to the results obtained on HMGB1-KO cells, SNS-Seq analysis of H1-TKO mES cells showed that reducing the histone H1 content in chromatin has a profound impact in the replication initiation landscape (Fig. 2a and Supplementary Fig. 2c). Although the most prominent SNS enrichments are still detectable, the widespread accumulation of short replication intermediates prevents a reliable application of any peak-calling algorithm to identify preferential sites of replication initiation in these cells (Supplementary Fig. 2d).

Concomitant analysis of replication dynamics on DNA fibres showed reduced interorigin distances and decreased fork speed in H1-TKO cells (Fig. 2b–c and Supplementary Fig. 1g-h and j). Most importantly, fork stability is strongly challenged in this condition, as indicated by the elevated number of stalled forks (Fig. 2d and Supplementary Fig.1i-j), and the increase in S-phase

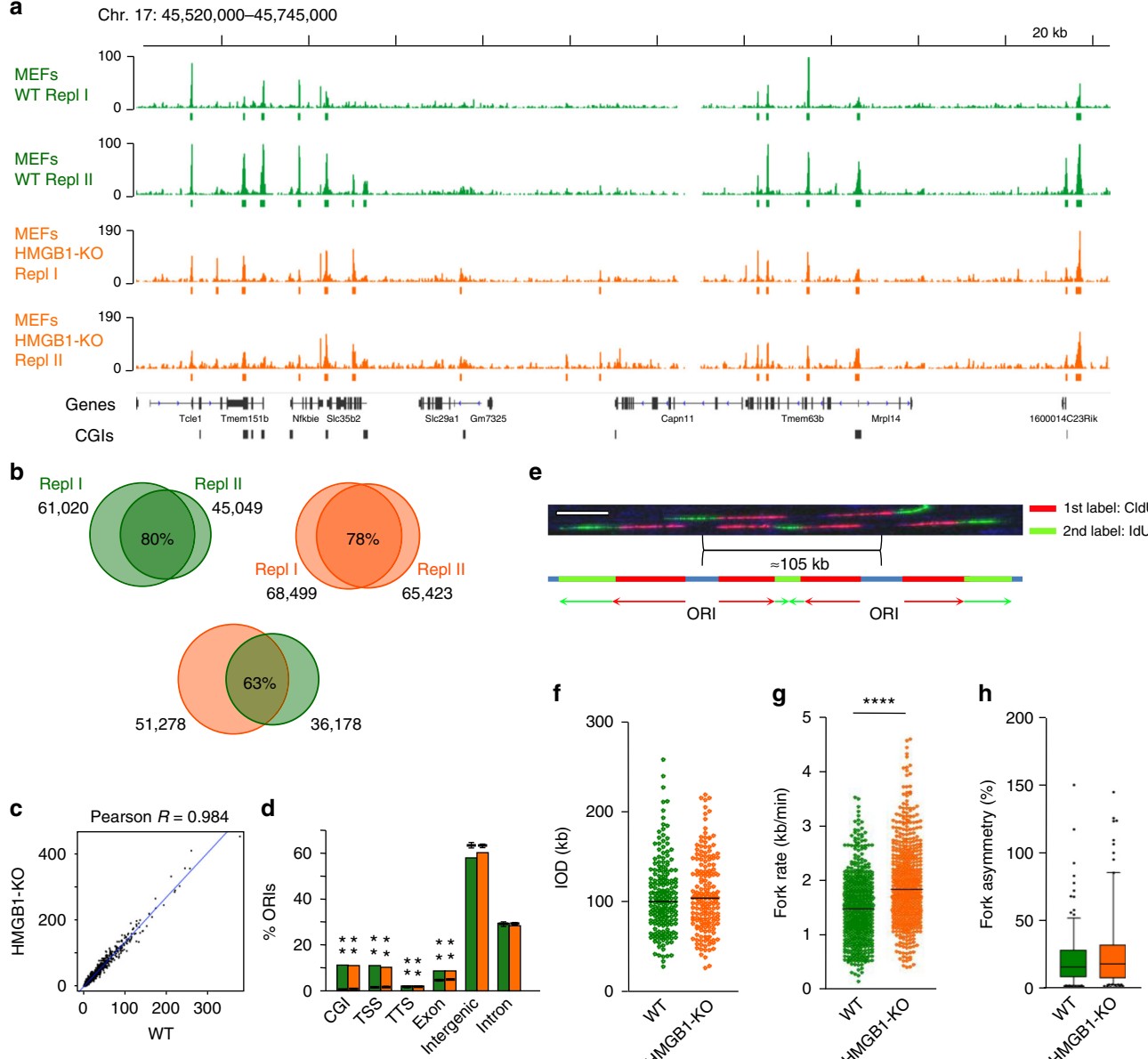

**Fig. 1** Replication initiation landscape and dynamics in cells with reduced nucleosome occupancies. **a** Representative IGV snapshot showing the SNS-Seq coverage and ORI locations from two biological replicates derived from 300 to 1500 nt SNS purified from wild-type (WT) MEFs (green tracks, upper two rows) or HMGB1-KO MEF cells (orange tracks, lower two rows). Colour rectangles below each track mark the position of the identified ORIs in each replicate. An additional representative genomic region is shown in Fig. 2a. **b** Venn diagrams showing the overlaps of common ORIs identified between replicates in WT (green) or HMGB1-KO cells (orange) and between both genotypes. **c** Pair-wise correlation of ORI number per genomic segment between WT and HMGB1-KO MEFs. **d** ORI enrichments at annotated genomic features in each cell type. Significances were calculated relative to randomized controls (depicted by black lines). The *P* value denotes the proportion of the randomization distribution that is larger than the observed test statistic. **$P <$ 0.001. **e** Representative example of DNA fibres labelled sequentially for 20 min with CldU (red) and IdU (green) used to estimate inter-origin distances (IOD), fork rates and fork asymmetries. Scale bar, 10 μm. **f** IODs, **g** fork rates and **h** percentage of fork asymmetry calculated from stretched fibres as those shown in **e**. Median values are indicated. Data are pooled from three replicate experiments (*n* = 3). Fork asymmetry is expressed as the ratio of the longest distance covered to the shortest, for each pair of sister replication forks. Box-plots spans the interquartile range and the inside segment show the median value. Whiskers above and below the box show the locations of the 90th and 10th percentile, respectively. Data not included between the whiskers are plotted as outliers (dots). Differences between distributions were assessed with the Mann–Whitney rank sum test. ****$P <$ 0.0001. See Supplementary Fig. 1b-e for full data analysis and numerical values

arrested cells (Supplementary Fig. 3d-f). A possible explanation for these results is that, when cells experience a shortage of H1, forks are slowed down and dormant ORIs are fired as a compensatory mechanism[30–33]. It is also possible that the altered chromatin context favours the activation of extra ORIs, and forks slow down due to a transient shortage of dNTPs, as described in other scenarios[34,35]. In agreement with augmented ORI activity,

we found higher MCM2 phosphorylation levels in H1-TKO cells, despite the reduced recovery of chromatin-associated proteins obtained from these cells by conventional cell-fractionation methods (Fig. 2e and Supplementary Fig. 4a). As previously reported, DNA damage signalling measured by CHK1 and H2AX phosphorylation were enhanced in this context of altered chromatin structure[28].

To get further understanding in how reduced histone H1 levels causes replicative stress, we first partially inhibit ORI firing by incubating the cells with the CDC7 kinase specific inhibitor PHA-767491 for 100 min, and analyse the effect on replication dynamics. Decreasing ORI activity (as shown by reduced levels of P-MCM2; Supplementary Fig. 4a), results on increased IODs and faster fork progression both in WT and in H1-TKO cells (Fig. 2f–g, PHA plots, and Supplementary Fig. 4b). This suggests

that cells compensate for the impaired ORI activity by increasing fork speed in both cell types. A similar effect on replication dynamics on both cell types was found upon supplementing the growth medium with nucleotide precursors (Fig. 2f–g, RbNs plots, and Supplementary Fig. 4b). These findings are in agreement with previous reports demonstrating that nucleoside supplementation can rescue fork progression even in situations of no detectable nucleotide depletion[36,37]. Interestingly, increasing

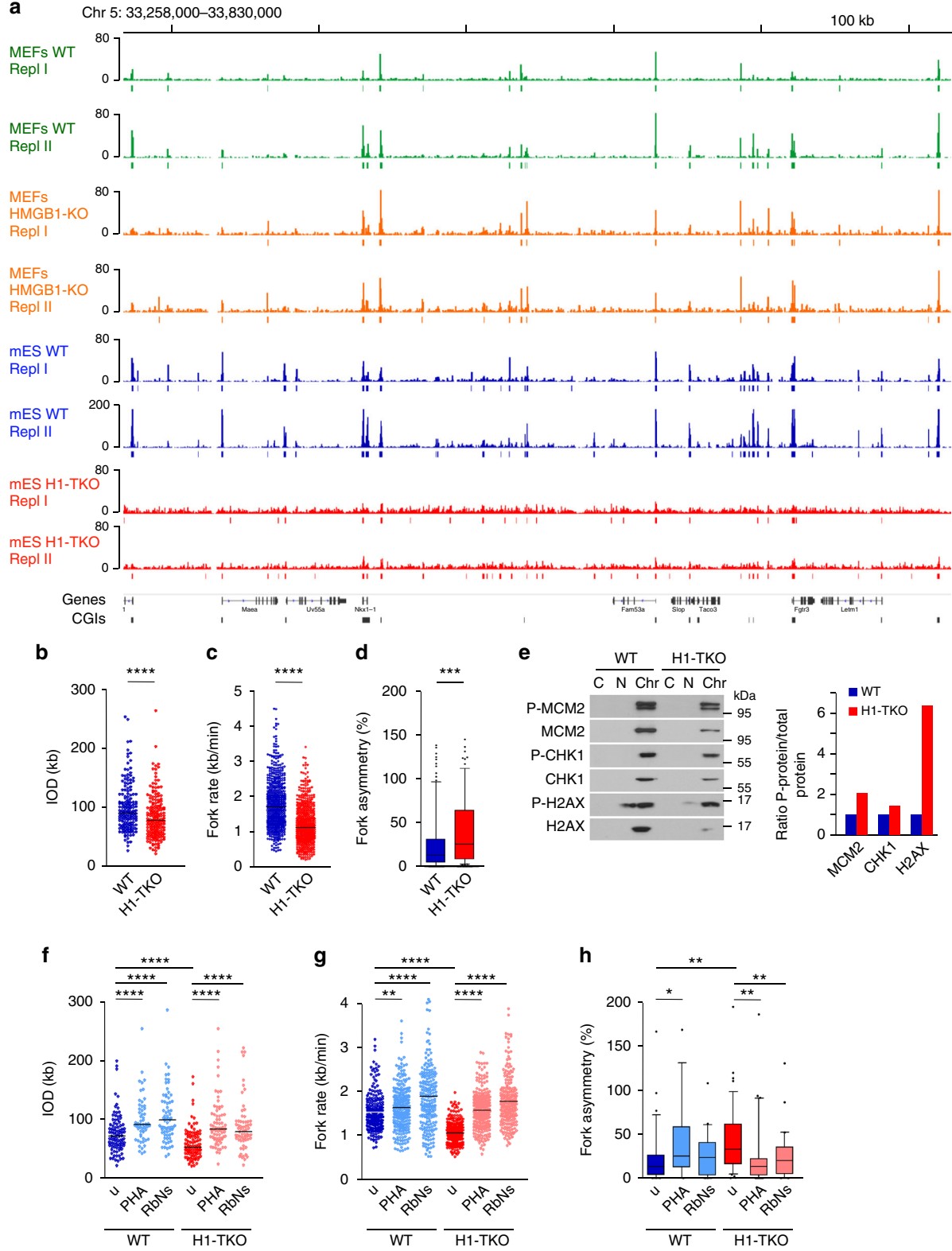

fork velocities by either mechanism relieves fork stalling in H1-TKO cells, suggesting that faster replisome speeds can alleviate the instability of elongating forks in H1-depleted chromatin (Fig. 2h and Supplementary Fig. 4b). Finally, to discard that some of the replication defects observed could be caused by indirect changes to the expression of DNA replication or cell cycle genes due to histone H1 reduction, we re-evaluated previously published RNAseq data on the same cells[29]. GO-term analysis failed to identify any significant enrichment in these term classes within the genes differentially expressed in H1-TKO cells (Supplementary Fig. 4c-d). Taken together, these analysis indicate that the strong alteration of the SNS landscape detected in H1-TKO cells reflects the firing of extra ORIs as well as a widespread accumulation of stalled replication intermediates.

**Replication timing is globally maintained in H1-TKO cells.** We next asked whether the replicative defects of H1-depleted cells underlay alterations in the temporal order of replication of their genome. For this we profile replication timing by microarray hybridization of early and late-replicating DNA derived from FACS-sorted cells[38]. We found that the segmentation in early and late-replicating domains was globally conserved between WT and H1-TKO cells, and similar to the timing profile of mES cells determined previously (Fig. 3a and Supplementary Fig. 5a)[39]. These results are in excellent agreement with the reported absence of global changes in the three-dimensional organization of the genome of these cells[29], as expected due to the strong correlation between replication timing domains and the genome's topological domains[40]. Around 3–4% of the genome in H1-TKO cells showed significant changes in replication timing relative to their matched WT cells, with a total of 59 and 38 regions (0.8–1 Mb in size) displaying delayed or advanced replication, respectively (Fig. 3b and Supplementary Data 1). Nearly 10% of the delayed regions overlap with the replication domains that undergo timing changes during mES cell differentiation towards neural precursors (Supplementary Figure 5b;[41]). Interestingly, both advanced and delayed regions displayed a signature of early replication domains such as higher CG content, gene-density and ORI density (Fig. 3c–e). Regions of altered replication timing in H1-TKO cells also showed cooperative loss or gain of chromatin marks such as H3K4me1 and/or H3K4me3, as reported for the altered topological compartments in histone H1-depleted cells (Supplementary Fig. 5c;[29]). Our analysis build up on the conclusion proposed by Geeven and collaborators stating that reduced levels of histone H1 results in epigenetic alterations at the most active chromosomal domains, including replication timing changes. Although we cannot discard that the limited number of shifts in replication timing could contribute to the altered replication profile of H1-TKO cells, they are likely not responsible for the genome-wide accumulation of replication intermediates and stalled forks (Fig. 2a–d).

**RNAPII dynamics is impaired in histone H1-depleted cells.** As one of the causes of fork stalling through the genome is transcription–replication encounters (reviewed in refs. [42,43], we hypothesized that the accumulation of stalled replication intermediates in H1-TKO cells was due to impaired RNAPII dynamics when chromatin contains reduced amounts of histone H1. To test this hypothesis we first investigated the effect of H1 depletion on the RNAPII elongation rate of two long genes, mediator complex subunit 13-like (*Med13l*) and inositol polyphosphate-5-phosphatase (*Inpp5α*), previously used to study RNAPII elongation dynamics in mouse cells[44]. Initiating RNAPIIs were transiently inhibited with 5,6-dichlorobenzimidazole1-β-D-ribofuranoside (DRB) and newly started primary transcripts were measured by qPCR spanning different exon–intron junctions across *Med13l* and *Inpp5α* gene bodies to detect synchronous pre-mRNA expression (Fig. 4a and Supplementary Table 1)[45,46]. Pre-mRNA transcription was readily detected at exon 1 of both genes 10 min after DRB removal, both in WT and in H1-TKO cells (light blue lines, Fig. 4b). However, the kinetics of transcription of the other exons differ in H1-TKO cells relative to WT cells, with pre-mRNA expression starting to be detected from earlier time points in mutant cells. Strikingly, expression of exon 5 at both genes (orange lines, Fig. 4b) was detected either at similar time points (in the case of *Med13l*), or much earlier (in the case of *Inpp5α*) than expression of exon 4 (red lines, Fig. 4b). These results suggest that, in conditions of H1 depletion, RNAPII initiates transcription not only from promoter–proximal positions but also within the body of the genes. Quantification of global nascent transcription by 5-ethynyl uridine (EU) incorporation following DRB release showed the enhanced recovery rate of RNA synthesis in H1-TKO cells, consistent with increased RNAPII initiation activity in conditions of reduced histone H1 levels (Fig. 4c and Supplementary Fig. 6a). Since previous work reported the absence of major differences in mRNA abundance between WT and H1-TKO cells[10,28,29], these results imply that the lack of H1 likely enhances non-productive transcription initiation.

As a second diagnostic test of altered RNAPII dynamics in cells with reduced H1 levels, we studied R-loop retention in chromatin by immunostaining with S9.6 antibodies[47]. R-loops are three stranded structures formed transiently during the transcription cycle when the nascent RNA molecule threads back and anneals to its template DNA strand, displacing the complementary DNA strand[48–50]. R-loops have been detected at many CpG island promoter regions and at the 3′-UTR of genes, where they seem to facilitate transcription termination[51]. However, their abnormal accumulation in chromatin has been correlated with genome instability[52–54]. We found that H1-TKO cells displayed significantly increased nuclear immunoreactivity than their WT counterparts (Fig. 4d and Supplementary Fig. 6b-c). Treatment of cells with RNAseA before immunostaining decreased most of cytoplasmic signal and some of the nuclear signal, while RNAseH treatment strongly reduced nuclear fluorescence, demonstrating

**Fig. 2** Replication initiation landscape and dynamics in cells with 50% reduction in histone H1 levels. **a** Representative IGV snapshot showing the SNS-Seq coverage and identified ORIs from two biological replicates derived from 300 to 1500 nt SNS purified from wild-type (WT) mES (blue tracks) or H1-TKO mES cells (red tracks). Data from MEFs WT and HMGB1-KO SNS-Seq are also shown for comparison (green and orange tracks). **b** IODs, **c** fork rates and **d** percentage of fork asymmetry calculated from stretched fibres from WT and H1-TKO cells. See Supplementary Fig. 1g-j for full data analysis and numerical values. Data are pooled from three replicate experiments ($n = 3$). Box-plots definitions are as in Fig. 1h. **e** Protein levels of P-MCM2, MCM2, P-CHK1, CHK1, P-H2AX and H2AX at the cytosolic (C), nucleoplasmic (N) and chromatin (Chr) fraction of both cell types. The quantification of the phosphorylated band intensity normalized to the corresponding total band intensity for each sample is shown on the right. **f** IODs, **g** fork rates and **h** % of fork asymmetry upon treating both cell types with PHA-767491 (PHA) or ribonucleosides (RbNs) for 100 min. Median values are indicated ($n = 2$). Differences between distributions were assessed with the Mann–Whitney rank sum test. ****$P < 0.0001$; ***$P < 0.001$; **$P < 0.01$; *$P < 0.05$. See Supplementary Fig. 4b for numerical values

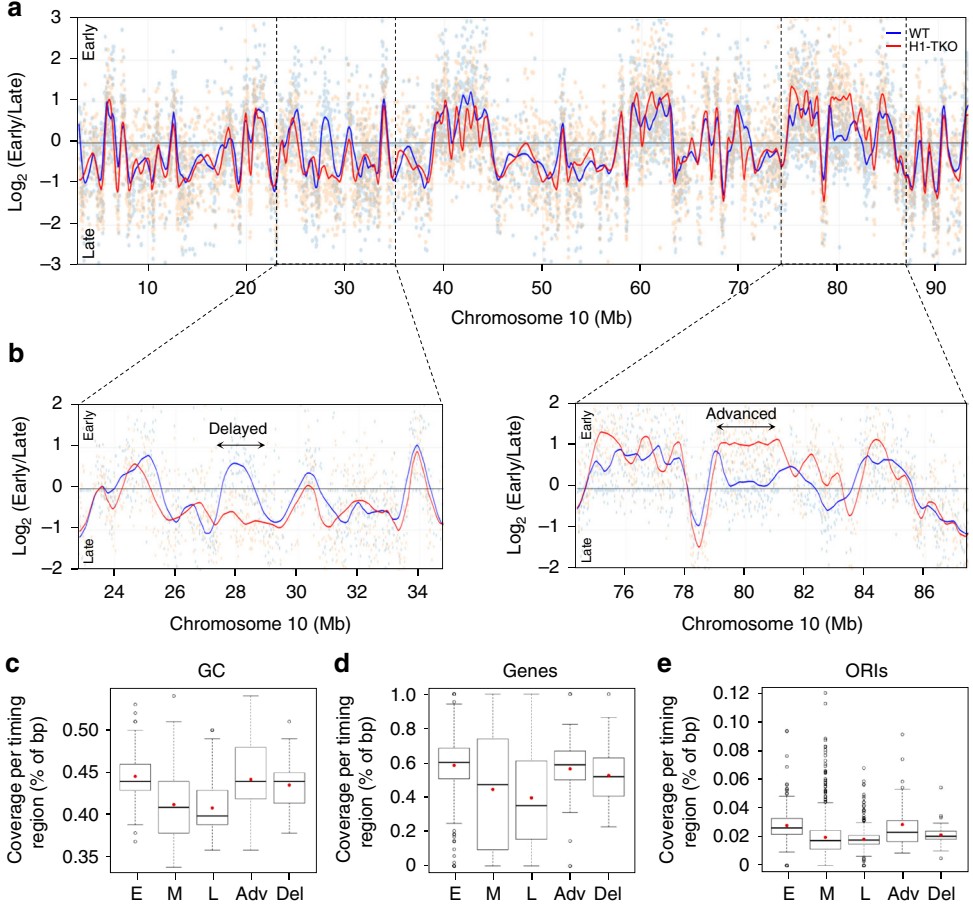

**Fig. 3** Replication-timing profiling of WT and H1-TKO mES cells. **a** Exemplary replication-timing profile of a large segment of Chromosome 10. Raw values for probe log ratios [$\log_2$(Early/Late)] and the local polynomial Loess smoothing curve from two replicate experiments of mES WT (blue) and mES H1-TKO (red) are shown. **b** Enhanced view of the two regions marked in **a** showing an exemplary delayed and advanced timing region in H1-TKO cells. See Supplementary Data 1 for the genomic location and features of the 97 regions with altered replication timing in H1-TKO cells. Coverage analysis of GC content (**c**), gene density (**d**) and ORIs (**e**) in the early (E), middle (M), late (L), advanced (Adv) and delayed (Del) replication-timing domains (% of bp). Box-plots definitions are as in Fig. 1h. Median values are indicated by a black line and means by a red dot. Data not included between the whiskers are plotted as outliers (empty dots)

that the increased antibody reactivity of H1-TKO cells was due to R-loops. We then used S9.6 immunoprecipitation of DNA/RNA hybrids followed by quantitative real-time PCR (DRIP-qPCR) to confirm R-loop accumulation at specific loci (Fig. 4e and Supplementary Table 1). We found a moderate, although similarly increased enrichment at most interrogated regions in H1-TKO cells relative to WT (Fig. 4e, A-histograms), including at the two genes displaying maximal differential expression in this cells (*Meg3* and *Rian*;[27,28]), at the two genes displaying altered transcription elongation dynamics (*Med13l* and *Inpp5a*), and at a control intergenic region that is not R-loop-associated in WT ES cells. Note that RNAseA-treated samples were used as input in the DRIP experiments to avoid free RNA species annealing to DNA during sample preparation (A-histograms). As in the immuno-fluorescence experiments shown in Fig. 4d, RNAseH treatment confirmed that DRIP specifically detected R-loops (Fig. 4e, H- and A + H-histograms). These data support that R-loop accumulation occurs throughout many genomic regions when histone H1 levels are reduced. Increased R-loop accumulation occurs mainly in early-S cells, in parallel with elevated levels of phosphorylated histone H2AX ($\gamma$H2AX) (Supplementary Fig. 6c-d), and was not detected in HMGB1-KO cells (Supplementary Fig. 6e), strongly indicating that deregulated R-loop resolution and concomitant

DNA damage signalling during S-phase are specific for histone H1-mediated chromatin alterations.

**RNAPII inhibition rescues H1-TKO replication stress**. Based on the range of alterations in RNAPII dynamics occurring in cells with reduced H1 content, we next asked whether the replication defects of these cells could be rescued by inhibiting RNAPII activity. Cells were incubated with $\alpha$-amanitin for 6 h, conditions in which RNAPII nascent transcripts were undetectable (Fig. 5a and Supplementary Fig. 7a). As expected for co-transcriptional R-loop formation, $\alpha$-amanitin-treated cells showed a significant reduction in S9.6 immunoreactivity, although this reduction did not reach the low signal levels detected when cells were incubated with RNAseH (Fig. 5b and Supplementary Fig. 7b). Importantly, analysis of replication fork dynamics by DNA fibres indicated that following transcriptional inhibition, both fork speed and fork stalling in H1-TKO cells displayed a clear recovery towards WT levels, whereas inter-origin distances were not significantly changed (pink plots, Fig. 5c–e and Supplementary Fig. 7c). These results suggest that encounters between replication and tran-scription complexes are pervasive in the genome of H1-TKO cells and a main cause of their altered replication pattern. RNAPII inhibition significantly reduced $\gamma$H2AX signal intensity both in WT and in H1-TKO mES cells (Fig. 5f and Supplementary

Fig. 7d), further confirming that clashes between replication and transcription machineries are a source of fork stalling and DNA damage in cycling cells.

Specific inhibition of early stages of the RNAPII transcription cycle with DRB rescued all the replication defects of H1-TKO cells (Fig. 6a–d, DRB). Most strikingly, this recovery was immediately reversed upon transcription re-start (Fig. 6a–d, DRB-release, and Supplementary Fig. 7e), strongly indicating that increased replication–transcription encounters are responsible for the replicative stress of cells with reduced H1 content. To note, the suppression of the replication defects of H1-TKO cells by DRB occurs with a milder decrease in P-MCM2 levels relative to

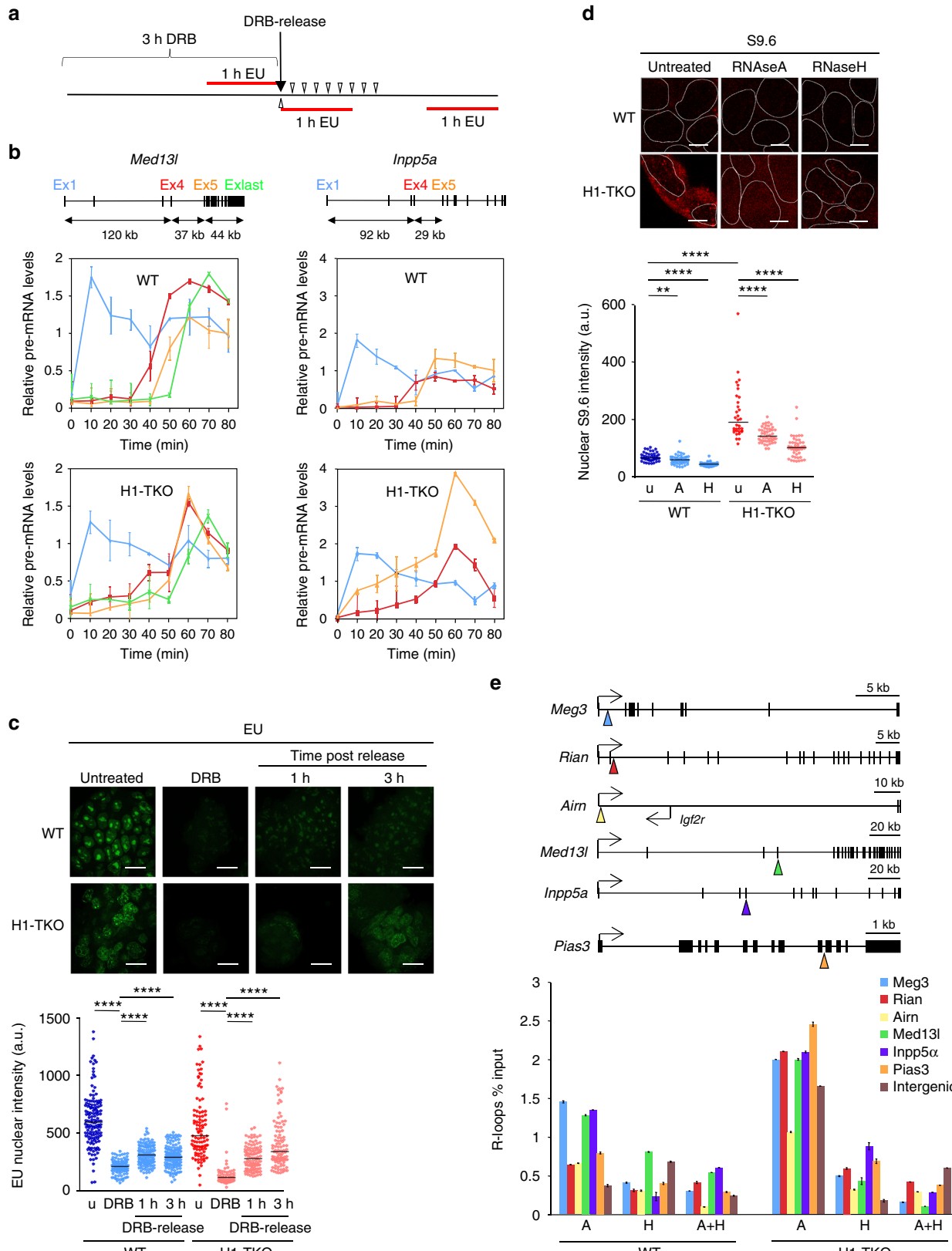

those obtained through CDK inhibition or RbNs supplementation (Supplementary Fig. 4a). Since RNAPII inhibition is the only of the assayed treatments that fully rescues the defects of H1-TKO cells without modifying fork dynamics in WT cells, taken together these results seem to suggest that increased ORI activation is a response to the replicative stress caused by replication–transcription conflicts in the absence of correct histone H1 levels. To evaluate the contribution of R-loop accumulation to the replication phenotype, we next performed transient transfection experiments to degrade DNA/RNA hybrids in vivo by overexpressing human RNAseH1 protein (Fig. 6e). RNaseH1 overexpression decreases both R-loop and γH2AX signalling (Fig. 6f–g and Supplementary Fig. 7f), and significantly enhanced the rate of replication fork progression, without fully recovering fork asymmetry (Fig. 6h–j and Supplementary Fig. 7g). Although it is possible that the activity of human RNAseH1 might not be sufficient to reverse all effects of accumulated R-loops in these cells, these results support that the observed replication impairment of H1-TKO cells is promoted by the presence of R-loops.

Altogether, these results demonstrate that the high levels of replication fork stalling occurring in H1-TKO cells account for increased transcription–replication conflicts due to altered RNAPII activity when histone H1 levels on chromatin are reduced. As the suppression of fork stalling upon RNAPII inhibition occurs concomitantly with the decrease of γH2AX signalling (Fig. 5f), the reported constitutively induced DNA damage checkpoint of these cells is likely due to persistent replicative stress in their division cycle[28].

**Histone reductions enhance transcription–replication rates**. A prediction from the above results is that in HMGB1-KO cells, where replication forks travel faster but no fork stalling was detected, RNAPII dynamics would not be altered. Time-course transcription elongation measurements upon DRB release at the *Med13l* and *Inpp5α* genes showed little changes in the timing of expression in HMGB1-KO cells relative to WT cells (Fig. 7a). Similarly, no alterations in R-loop resolution or in γH2AX signalling were found in cells with reduced nucleosome numbers (Supplementary Fig. 6e). These data suggest that cells can tolerate higher replication velocities as long as RNAPII dynamics is not impaired.

We next tested the complementary scenario: are replication forks affected in situations where reduced histone content causes increased RNAPII elongation rates? To address this issue, we analysed a human cell line whose levels of canonical histones can be modulated by knocking down the expression of the stem–loop-binding protein (SLBP) gene, encoding a histone mRNA regulatory factor[55] Upon mild reduction of SLBP levels

(Supplementary Fig. 8a and[46], no cell-cycle defects were observed, but the amount of canonical histones and variants H2A.X and H2A.Z on chromatin decreased, causing an increase in RNAPII elongation rates and consequent defects in co-transcriptional splicing[46]. DNA fibre analysis of SLBP-knockdown cells showed a strong increase in fork speed, without significant changes in inter-origin distances or fork asymmetry (Fig. 7b–d and Supplementary Fig. 8b). Thus, the rate of replisome progression in vivo can vary over a range of velocities without compromising fork stability. Indeed, the average increase in fork velocity was stronger in this situation of histone depletion than in HMGB1-KO cells (40% vs. 29%) (Fig. 7e), consistent with the increased RNAPII elongation rates displayed by SLBP-knockdown cells[46]. Possibly, this enhanced response on both transcription and replication elongation rates could be associated with the changes in the composition of chromatin occurring specifically in SLBP-knockdown cells, such as the drop in the levels of the histone variant H2A.Z and the increase of the variant H3.3[14,46].

Collectively, these results illustrate that a more open chromatin conformation due to fewer nucleosomes enable faster progression of both replication and transcription machineries to different extents, and suggest that this co-regulation can act as a mechanism to diminish the probability of conflicts between both complexes helping cells to avoid replicative stress.

## Discussion

The results reported here support a scenario by which perturbations in the integrity of the chromatin template elicit a range of responses in the dynamics of DNA replication and transcription, with different consequences on replicative stress. We found that reductions of the histone-DNA ratios in chromatin, either resulting from impaired histone incorporation by knocking-out HMGB1, or from mild alterations in the histone supply by knocking-down SLBP, allowed faster fork velocities without altering the global replication initiation landscape or generating fork instability (Figs. 1 and 7). These findings were unanticipated, as fork velocities and ORI usage appear to be coupled in all cellular contexts examined[30–34,56] and likely reflects the cells' flexibility to cope with variations in fork speed providing that no replication factor becomes limiting. Replication fork speed in vivo is dependent on new histone supply and efficient nucleosome reassembly (reviewed in ref. [57]) Indeed, long-term inhibition of canonical histone biosynthesis resulting in a much more severe reduction in histone levels in chromatin that the ones reported here, strongly impaired replication fork progression and generated DNA damage[58,59]. In turn, lowering nucleosome occupancies in chromatin accelerated RNAPII elongation rates to various degrees (Fig. 7 and ref. [46]). These results indicate that nucleosomes exert a physical barrier to the progression of the

**Fig. 4** H1-TKO cells display impaired transcription dynamics. **a** Diagram of the experimental design to measure transcription elongation rates by transient inhibition of initiating RNAPIIs with DRB. Three hours after DRB incubation, the drug was washed-off to resume transcription elongation and total RNA was extracted from identical number of cells at the indicated time-points (open triangles). Global nascent transcription was evaluated by 1h-EU labelling at the indicated time points (red lines). **b** Time course transcription elongation measurements at the *Med13l* and *Inpp5a* genes in WT mES (upper panels) and H1-TKO mES cells (lower panels). Levels of pre-mRNA at the indicated times were determined by RT-qPCR at the positions marked in the gene maps above the graphs. Pre-mRNA values were normalized to the values of the non-DRB-treated sample. Results are shown as means ± s.d. from two independent experiments (*n* = 2). **c** Representative images of EU staining (top) and distribution of EU nuclear intensity during DRB treatment and upon drug-release at the time points shown in the experimental scheme in **a** (bottom). Scale bar, 20 μm. Statistical analyses and normalized values to those obtained at untreated cells are shown in Supplementary Fig. 6a. **d** Representative images of S9.6 immunostaining ±RNAseA or +-RNAseH incubation (top) and distribution of S9.6 nuclear intensity (bottom) in WT and H1-TKO cells. Scale bar, 10 μm. Nuclear segmentation (white lines) was based on DAPI staining. Median values are indicated (*n* = 2). See Supplementary Fig. 6b for numerical values and Supplementary Fig. 6c-d for S-phase distribution of S9.6 and γH2AX nuclear intensities. Differences between distributions were assessed with the Mann–Whitney rank sum test. ****$P < 0.0001$; ***$P < 0.001$; **$P < 0.01$. **e** DRIP analysis of R-loop enrichment at the indicated genes and at an intergenic control region. Primer positions are shown by colour triangles on the gene maps. Values are % of input from samples treated with RNAseA (A), RNAseH (H) or both (A + H). Means ± s.d. from two independent experiments are shown (*n* = 2). See Supplementary Table 1 for primer pair sequences and qPCR conditions

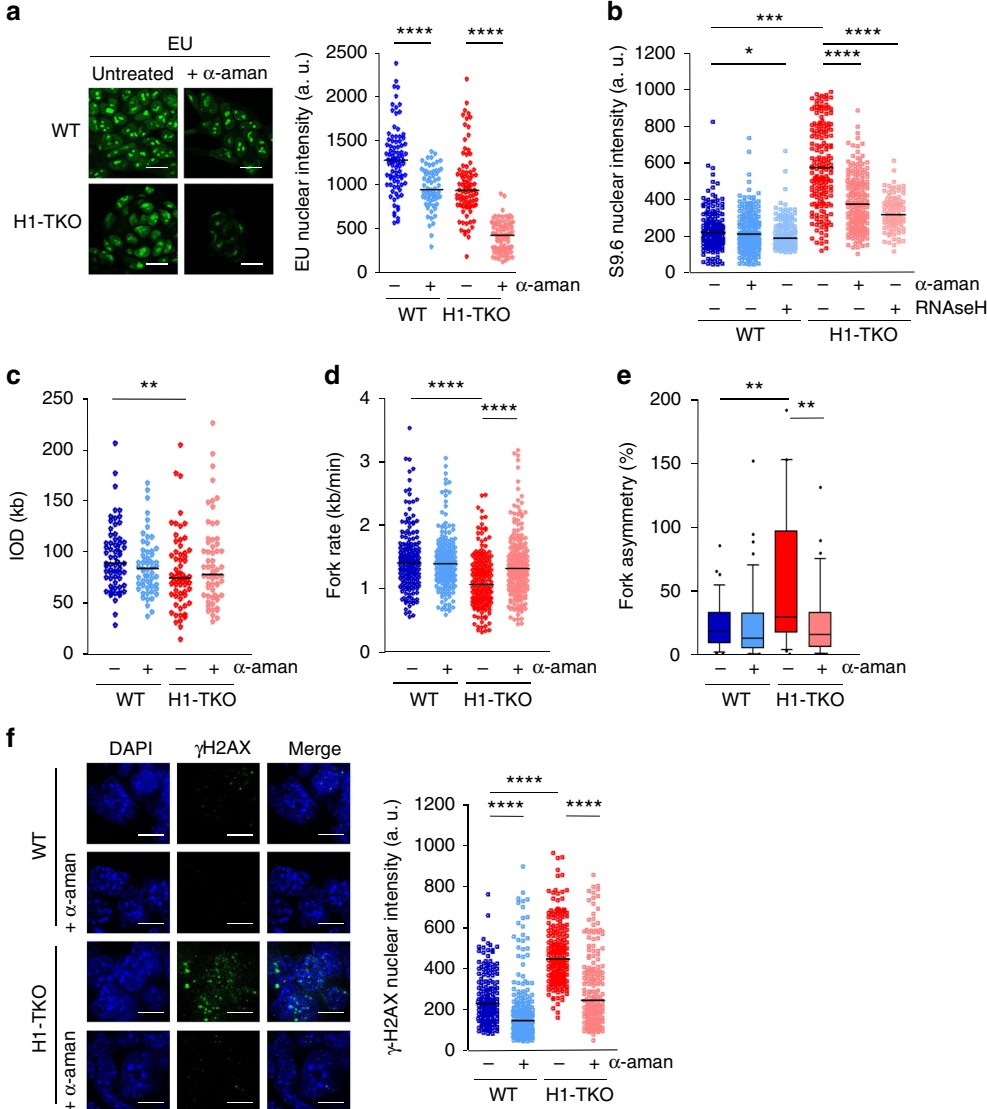

**Fig. 5** RNAPII inhibition suppresses the replication phenotypes of H1-TKO cells. **a** Representative images and quantification of nuclear EU staining of WT and H1-TKO cells untreated or treated with *a*-amanitin for 6 h. Scale bar, 20 μm. **b** Distribution of S9.6 nuclear intensity ±RNAseH incubation in WT mES and H1-TKO mES cells untreated or treated with *a*-amanitin. Data are pooled from two independent experiments (*n* = 2). Median values are indicated. **c** IODs, **d** fork rates and **e** % of fork asymmetry calculated from stretched fibres of cells untreated or treated with *a*-amanitin. **f** Representative images of γH2AX foci (left) and distribution of γH2AX intensity per nucleus (right) in cells untreated or treated with *a*-amanitin. Scale bar, 10 μm. Median values are indicated. Data are pooled from two independent experiments (*n* = 2). Differences between distributions were assessed with the Mann–Whitney rank sum test. ****$P < 0.0001$; ***$P < 0.001$; **$P < 0.01$; *$P < 0.05$. See Supplementary Fig. 7a-d for numerical values

molecular machines that copy the genetic information and suggest that ongoing replisomes and elongating RNAPII complexes might have different requirements for remodelling activities to destabilize nucleosomes ahead of them. Importantly, we found that cells can tolerate a range of variations in the speed of replication and transcription elongation complexes without experiencing detectable problems in their division cycle as far as these increased rates do not generate collisions between both machineries. The opposite situation is exemplified by the reduction of histone H1 levels in chromatin, which triggers a dramatic alteration in replication initiation patterns, including the firing of additional ORIs and massive fork stalling as results from elevated transcription–replication conflicts (Figs. 2, 5 and 6). In this altered chromatin context, surprisingly, we observed that fork instability can be alleviated by increasing the rate of replication fork progression. The reasons for this are unclear, but we speculate that slow-moving forks in H1-depleted chromatin are specifically

sensitized to encounters with transcription roadblocks such as paused or arrested RNAPII complexes or accumulated R-loops.

An unexpected finding of this work was that, in the absence of the correct amounts of histone H1 in chromatin, the dynamics of RNAPII is defective, inducing replication stress and DNA damage signalling (Figs. 4 and 5). These findings reveal a role of histone H1 in regulating the initial steps of the RNAPII transcription cycle and/or in R-loop resolution whose mechanistic details deserve further investigation. Supporting our results, a recent report demonstrates that *Drosophila* histone H1 prevent R-loop-induced replication stress in heterochromatin[60]. While depletion of the only somatic histone H1 variant in *Drosophila* induces abnormally early heterochromatin replication[60] we found that the temporal replication programme of cells with a 50% reduction in histone H1 levels was globally preserved, with only few regions shifting replication timing (Fig. 3). Although we cannot rule out that heterochromatic regions like telomeres or centromeres could

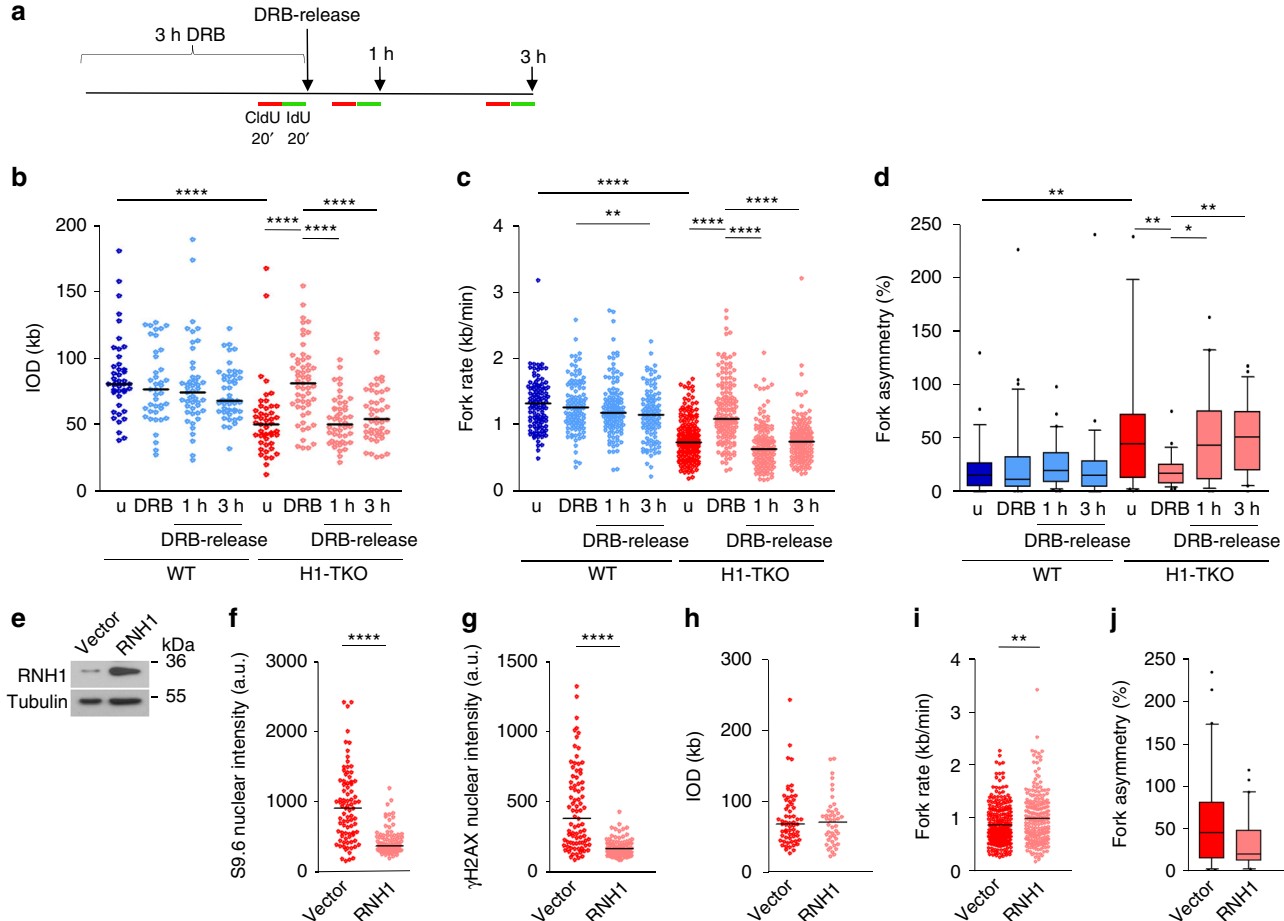

**Fig. 6** Suppression of the replicative stress of H1-TKO cells by RNAPII inhibition is immediately reverted upon transcription re-start. **a** Diagram of the experimental design to measure replication dynamics by transient inhibition of initiating RNAPIIs with DRB. Cells were labelled sequentially for 20 min with CldU (red) and IdU (green) at the indicated time points before fibre stretching. **b** IODs, **c** fork rates and **d** % of fork asymmetry from WT (blue) and H1-TKO (red) cells untreated (u), treated with DRB for 3 h (DRB) and released from the DRB block (1 h and 3 h DRB-release) as shown in **a** ($n = 2$). **e** Protein levels of RNAse H1 (RNH1) and TUBULIN in H1-TKO cells transfected either with control (vector) or RNAseH1-overexpression vector (RNH1). **f**, **g** Distribution of S9.6 and γH2AX nuclear intensities in the same cells (vector, red plots; RNH1, pink plots). **h** IODs, **i** fork rates and **j** % of fork asymmetry of the same cells. Median values are indicated ($n = 2$). Differences between distributions were assessed with the Mann–Whitney rank sum test. ****$P <$ 0.0001; ***$P < 0.001$; **$P < 0.1$; *$P < 0.2$. See Supplementary Fig. 7e-g for numerical values. Box-plots definitions are as in Fig. 1h

also display altered replication timing in H1-TKO cells, as these regions were excluded from the microarrays used in the analysis, the absence of main changes in the late-replicating segments of the genome, as well as in the S-phase distribution of EdU patterns (Supplementary Fig. 3f), and in the global 3D topological organization[29], argues against this possibility. On the contrary, the widespread distribution of replication intermediates (Fig. 2a), and the rapid effect of transcription inhibition and re-start on replication fork dynamics (Figs. 5 and 6), strongly suggest that pervasive RNAPII roadblocks can, either directly, or indirectly through the generation of R-loops, interfere with the progression of the replisomes along the genome generating replicative stress and causing the firing of extra ORIs. Interestingly, a recent report demonstrated that replication progression itself modulate R-loop formation and resolution in an orientation-dependent manner[61]. Is therefore possible that the highly deregulated replication occurring in H1-TKO cells, with augmented ORI usage, can directly affect R-loop accumulation, in turn enhancing transcription–replication conflicts and replicative stress. Increasing evidence supports the notion that altered transcription and R-loop formation can contribute to replicative stress and drive genomic instability[37,62,63]. Accumulated R-loops can also mediate

chromatin compaction at certain genomic regions, contributing to fork stalling and aggravating replication-transcription conflicts at those sites[64,65]. We expand this scenario further by showing that impaired RNAPII metabolism caused by alterations in chromatin structure can be an additional mechanism triggering replication stress in cycling cells.

Our work shows that chromatin conformation regulates the dynamic interplay between replication and transcription, unveiling its fundamental role in minimizing conflicts between the machineries moving along the same DNA template, and thus in preserving genome integrity. These findings could have important implications in situations of physiologically or pathologically altered chromatin structure, such as cellular aging, neurological and developmental disorders, or cancer[66–69].

## Methods

**Cell culture, cytometry and drug treatment**. mES cells (kindly provided by A. Skoultchi, Albert Einstein College of Medicine, USA) were grown at 37 °C and 5% CO2 in Dulbecco's modified Eagle's medium (DMEM, Invitrogen) supplemented with 10% foetal calf serum, FBS (Biosera), 1× non-essential aminoacids, 2 mM L-glutamine, 50 mM b-mercaptoethanol, 1 mM sodium piruvate, 100 U/ml penicillin, 100 µg/ml streptomycin (Invitrogen) and 10³ U/ml LIF (ESGRO). MEFS were derived from 14.5 dpc (Balb/C) embryos and grown in DMEM high glucose

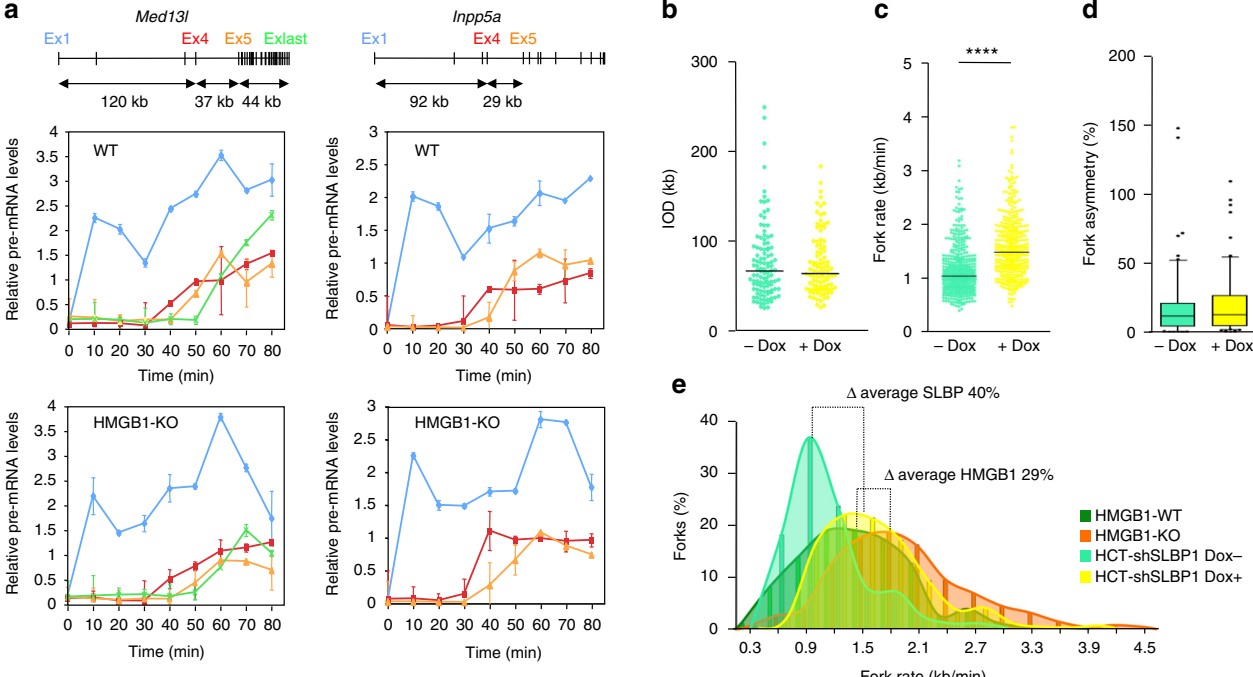

**Fig. 7** Reduced histone content allows faster rates of replication and transcription. **a** Time course transcription elongation measurements at the *Med13l* and *Inpp5a* genes in WT MEFs (upper panels) and HMGB1-KO MEFs cells (lower panels) following the experimental design depicted in Fig. 4a. Levels of pre-mRNA at the indicated times were determined by RT-qPCR at the positions marked in the gene maps above the graphs. Pre-mRNA values were normalized to the values of the non-DRB-treated sample. Results are shown as means ± s.d. from two independent experiments (n = 2). **b** IODs, **c** fork rates and **d** percentage of fork asymmetry calculated from stretched fibres of HCT-shSLBP.1 cells cultured in the absence (light green) or presence (yellow) of Doxicyclin (Dox) for 72 h. Median values are indicated. Data are representative of two replicate experiments and pooled (n = 2). Differences between distributions were assessed with the Mann–Whitney rank sum test. ****P < 0.0001. See Supplementary Fig. 8b for numerical values. Box-plots definitions are as in Fig. 1h. **e** Distribution of replication fork speeds in cells with reduced histone content relative to their respective WT counterparts

medium supplemented with 10% FBS (Gibco), 100 U/ml penicillin, 100 μg/ml streptomycin, 2 mM L-glutamine, 1× non-essential aminoacids and 50 mM β-mercaptoethanol. HTC-shSLBP.1 cells (kindly provided by F. Prado, CABIMER, Spain) were cultured with or without Dox for 3 days as described[46] in McCoy's 5A-modified medium supplemented with 10% FBS (Gibco), 100 U/ml penicillin and 100 μg/ml streptomycin. All cell lines have been tested for mycoplasma contamination. For cell-cycle analyses cells were incubated for 20 min with 250 μM IdU (Sigma) and fixed in 70% ethanol. After 30 min incubation in 2 M HCl with 0.5% Triton X-100 samples were neutralized for 2 min in 100 mM Sodium Tetraborate, pH 9.5, and then incubated for 1 h with primary antibodies (mouse anti-BrdU/IdU, BD Bioscience) and 30 min with secondary antibodies (anti-mouse IgG Alexa-Fluor 647, Thermo Fisher Scientific), both at room temperature. After that cells were stained with cycle buffer (PI/RNAse, BD Pharmingen) for 30 min at room temperature in the dark. Samples were analysed using a FACSCanto A machine (Becton Dickinson), FACSDiva v6.1.3 analysis software, and the FlowJo v10 programme.

RNAPII-dependent transcription was inhibited by incubating cells in the presence of either 100 mM DRB for 3 h (Sigma Aldrich) or 10 mg/ml α-amanitin for 6 h (Sigma Aldrich). CDC7 inhibition and ribonucleosides supplementation were performed by incubating cells with 60 μM of PHA-767491 (Sigma) or 60 μM of Nucleosides (Merck) for 100 min, respectively.

**DNA transfection**. For each experiment, 7.5 × 10^6 cells were co-transfected with a mixture of 1 μg mCherry reporter plasmid + 4 μg pcDNA3.1 vector or 4 μg pcDNA3-RNAseH1 vector (gift from A. Aguilera, CABIMER, Spain) using Lipofectamine 2000 reagent (Invitrogen). Twenty four hours after transfection, 0.5 × 10^6 mCherry-positive cells were sorted (FACSAria Fusion, BD Biosciences), and grown for another 24 h before fibre stretching and immunofluorescence analysis.

**Short nascent strands purification**. For each SNS preparation, approximately 10^9 exponentially growing cells were directly lysed on the flasks with 50 mM Tris pH 8.0; 10 mM NaCl; 10 mM EDTA pH 8.0; 0.5% SDS and 0.1 mg/ml of Proteinase K (Roche) and incubated overnight at 37 °C in a water bath. High molecular-weight genomic DNA was purified by extremely careful phenol/chloroform extractions and precipitated with 0.1 volumes of 3 M sodium acetate and 2 volumes of cold absolute ethanol in a 50 ml tube. DNA skeins were fished with a Pasteur pipette

and gently rinsed three times with 1 ml 70% ethanol by sequential transfer into fresh eppendorf tubes. Once dried, DNA pellets were resuspended in 1 ml of TE supplemented with 2 μl of RNAseOUT (Thermo Fisher Scientific) and kept at 4 °C for 48–96 h.

Sucrose gradients were prepared from seven-step sucrose solutions (from 5% to 20% sucrose) made in 10 mM Tris-ClH (pH 8.0), 1 mM EDTA, 100 mM NaCl in centrifuge tubes (Beckman Coulter 331374) using a peristaltic pump. Fully resuspended genomic DNA was denatured by heating during 5 min at 100 °C and immediately transferred on ice. Denatured gDNA was gently transferred on the top of each sucrose gradient tube and size fractionation was performed by ultracentrifugation on SW40Ti rotors during 20 h, at 102,445 × g and 20 °C in a Beckman Coulter Optima™ L-100 XP Ultracentrifuge. A total of 1 ml fractions were collected from the top of the gradient and precipitated by adding 1 μl of 20 mg/ml glycogen, 100 μl NaOAc 3 M pH 5.2 and 2.2 ml EtOH 100%. Air-dried replication intermediates from each gradient fraction were resuspended in 60 μl of TE supplemented with 1 μl of RNAseOUT™. Size fractionation was analysed by alkaline electrophoresis in 1% agarose gels (50 mM NaOH, 1 mM EDTA). Those fractions containing replication intermediates ranging between 400 and 2000 nt of size derived from four (mES) to eight (MEFS) independent SNS preparations were pooled. To avoid any possible contamination with nicked DNA generated during sample manipulation and thus enrich in replication intermediates (containing an RNA primer at their 5′-end), pooled fractions of the desired sizes were subject to three rounds of PNK phosphorylation and lambda-exonuclease digestion as follows. Samples were heat denatured during 5 min and the phosphorylation reaction was prepared in 200 μl with 1× forward buffer, 100 U PNK (Thermo Scientific), 1 mM ATP (Thermo Scientific), and 40 U RNAseOUT™ and incubated for 30 min at 37 °C. Reactions were stopped by adding 5 μl of 5% sarkosyl, 5 μl of 0.5 M EDTA and 10 μl of Proteinase K (625 μg/ml) and incubated for 30 min at 50 °C. Samples were extracted with phenol/chloroform, ethanol precipitated and resuspended in 100 μl of water. Once resuspended, they were heat denatured again for 5 min and the digestion with lambda-exonuclease was prepared in 150 μl with 1× buffer, 150 U lambda-exonuclease (custom-made, Thermo Scientific) and 40 U RNAseOUT™ and incubated overnight at 37 °C. Reactions were inactivated during 10 min at 75 °C, extracted with phenol/chloroform, ethanol precipitated and resuspended in 60 μl of water (or in 30 μl after the third round).

To prepare replication intermediates for sequencing, SNS were primed with 1 μl (50 pmol) of random hexamer primer-phosphate (Roche) by incubating the reactions for 5 min at 95 °C and allowing gradually cooling to 4 °C. Primer extension was performed by adding 1 μl dNTP (10 mM) (Roche) and 1 μl exo-Klenow 5 U/μl (New England Biolabs), and incubating for 1 h at 37 °C. Reactions were inactivated for 10 min at 75 °C. Ligation of adjacent fragments of the second-strand synthesis was performed by adding 2 μl of Taq DNA ligase 40 U/μl (New England Biolabs), incubating for 30 min at 50 °C and inactivating the enzyme for 10 min at 75 °C. Finally, RNA primers were removed by treating samples with 1 μl RNAse A/T1 Mix (Thermo Scientific) for 30 min at 37 °C, and DNA was extracted with phenol/chloroform, ethanol precipitated and resuspended in 20 μl of TE. DNA libraries were prepared at the Fundación Parque Científico de Madrid (FPCM), with the NBNext kit (New England Biolabs) following manufacturer's instructions. Each library was sequenced by 75 single-end runs on a NS500 system (Illumina) at the FPCM.

**SNS-Seq data processing**. SNS-Seq reads were aligned to the mm10 version of the genome using the standard BWA-MEM algorithm. Reads with low quality and multihits were removed with the Samtool view (parameter -q 1). ORI peaks were determined with the scanquantile programme[19], with a P value threshold of 1E −16. The required genome segmentation used by this peak-calling script was based on replication timing data from mES[39], which accurately matches the read coverage differences between segments. Finally, the peaks separated by less than 200 bp were merged (keeping the lowest P value), and the peaks with a P value higher than a threshold were removed; this threshold was fixed depending on the read depth and the background of each track.

Common peaks were obtained with the intersectBed (BedTools), with parameters -wa -f 0.1: nonreciprocal, and with a minimal fraction of 10% of overlap. The heatmap, pairwise comparison between origins per segment, and Venn diagrams were generated with custom R scripts; the dendrogram of the heatmap was calculated with the standard hierarchical clustering implemented on the heatmap.2 function (gplot package).

To account for the observed genomic distribution of replication origins, ORI peaks were compared with the expected proportion calculated from genomic intervals randomly sampled from throughout the genome. The procedure was repeated 1000 times and the statistical significance of the observed distribution was determined by computing the empirical P value from the sampling distribution.

**Fibre stretching**. Exponentially growing cells were pulsed first with 50 μM CldU (Sigma) for 20 min and then with 250 μM IdU (Sigma) for another 20 min. Cells were then resuspended at a concentration of $0.5 \times 10^6$ cells/ml in cold PBS. A volume of 2 μl of cell suspension were lysed directly on the slide by adding 10 μl of pre-warmed (30 °C) spreading buffer (0.5% SDS, 200 mM Tris pH 7.4, 50 mM EDTA) and incubated for 6 min at room temperature in humidity chamber. DNA fibres were stretched by tilting the slide ~30°. After air drying the slides, DNA fibres were fixed for 2 min with cold (−20 °C) 3:1 methanol:acetic acid solution. Slides were then incubated in 2.5 M HCl for 30 min at room temperature, washed three times with PBS, and blocked for 1 h with 1% BSA, 0.1% Triton X-100, 1× PBS before incubation with 1:100 anti-CldU (Abcam), 1:100 anti-IdU (BD Bioscience) and 1:300 anti-ssDNA (Millipore) antibodies during 1 h at room temperature in humidity chamber. Finally, the slides were incubated for 30 min at room temperature with 1:300 anti-rat IgG Alexa Fluor 594 (Thermo Fisher Scientific), anti-mouse IgG1 Alexa Fluor 488 (Thermo Fisher Scientific) and anti-mouse IgG2a Alexa Fluor 647 (Thermo Fisher Scientific), air dried and mounted with Prolong (Invitrogen). Visual acquisition of the DNA fibres was done in a Axiovert200 (Zeiss) Fluorescence Resonance Energy Transfer microscope and the images were analysed with the image processing programme ImageJ v1.51a using the conversion factor of 1 μm = 2.59 kb. Statistical analyses were performed in Prism v5.0.4 (GraphPad Software) using the non-parametric Mann–Whitney rank sum test . *P < 0.05, **P < 0.01, ***P < 0.001, ****P < 0.0001.

**Protein extraction and western blotting**. Cells were lysed with RIPA buffer (25 mM Tris-HCl pH 7.5, 180 mM NaCl, 1% deoxycholate, 1% NP-40, 0.1% SDS) and extracts were sonicated in a Bioruptor (Diagenode) during 10 cycles 30 s HIGH, 30 s OFF, and then cleared by centrifugation (15 min, 4 °C, 21,100 × g). Primary antibodies used were rabbit anti-P-MCM2-S53 (Abcam ab70367, dilution 1/1000), rabbit anti-MCM2 (gift from Juan Méndez, CNIO, Spain, dilution 1/5000), rabbit anti-P-CHK1-S345 (Cell Signalling, 2348 S, dilution 1/500), mouse anti-CHK1 (Santa Cruz sc-8404, dilution 1/500), rabbit anti-γH2AX (Cell Signalling 9718S, dilution 1/1000), rabbit anti-H2AX (Cell Signalling 7631S, dilution 1/1000), rabbit anti-RNAseH1 (Proteintech 15606-AP, dilution 1/2000), mouse anti-α-TUBULIN (Sigma T-5168, dilution 1/4000), rabbit anti-HMGB1 (Abcam ab18256, dilution 1/1000) and mouse anti-SLBP (Sigma WH0007884M1, dilution 1/500). Uncropped scans of the blots are shown in Supplementary Fig. 9.

**Replication timing analysis**. In total, $30 \times 10^6$ WT and H1-TKO MEF cells were incubated with BrdU for 90 min before ethanol fixation. After centrifugation, fixed cells were suspended in PBS with RNAse A (0.5 mg/ml) and propidium iodide (50 μg/ml) for 30 min at room temperature. 100,000 cells were sorted into two

fractions, S1 and S2, corresponding to Early and Late S-phase fractions, respectively, using a FACSAria Fusion apparatus (BD Biosciences). Cells in each fraction were then suspended in a lysis buffer (50 mM Tris pH = 8, 10 mM EDTA, 300 mM NaCl, 0.5% SDS, and 0.2 mg/ml of Proteinase K) and newly synthesized DNA was then immuno-precipitated with BrdU antibodies (Anti-BrdU Pure, BD Biosciences, 347580). The quality of enrichment of early and late fractions in S1 and S2 was performed by qPCR with Pou5f1 oligonucleotides (early control) and with Akt3 oligonucleotides (late control) as described previously[38]. Whole genome amplification was conducted (WGA, Sigma) to obtain the amount of DNA (1000 ng) required for microarray hybridization as recommended by the manufacturer. After amplification, early and late newly synthesized DNAs were labelled with Cy3 and Cy5 ULS molecules (Genomic DNA labelling 24Kit, Agilent) as recommended by the manufacturer. The hybridization was performed according to the manufacturer's instructions on 4 × 180K mouse microarrays (SurePrint G3 Mouse CGH Microarray Kit, 4 × 180K, Agilent Technologies, reference genome: mm9) that covers the whole genome with one probe every 13 Kb (11 Kb in RefSeq sequences). Microarrays were scanned with High-Resolution C Scanner (Agilent) using a 2 μm resolution and the autofocus option. Feature extraction was performed with the Feature Extraction 9.1 software (Agilent Technologies). Analysis was performed with the Agilent Genomic Workbench 5.0 software. The log 2 -ratio timing profiles were smoothed using the Agilent Genomic Workbench 5.0 software with the Triangular Moving Average option (500 kb windows). To determine the replication domains in different conditions, algorithms from CGH applications in the Agilent Genomic Workbench 5.0 software were used, particularly the aberration detection algorithms (Z-score with a threshold of 1.8) that define the boundaries and magnitudes of the regions of DNA loss or gain corresponding to the late and early replicating domains respectively. Then, a comparative analysis of replication domains was performed between the different cellular conditions, in order to determine DNA segments with significant replication timing changes. A Student's test (t-test) was performed on the average of the Log 2 values of every domains with R programme 3.2.3 and significant difference is annotated when P value < 10 $^{-3}$. The intersection with different data sets was performed with GALAXY tools and T-test was performed to identify significant differences. Positions of genes used for gene coverage come from RefSeq mm9. The characteristics of the regions with significant alteration in replication timing in H1-TKO cells are shown in Supplementary Data 1. To check the overlap between these regions and domains that change replication timing during differentiation of mES cells in vitro[69], the percentage of intersection between them was compared with the expected percentage obtained when randomizing 10,000 times their genomic distribution. Statistical significances were determined by computing the empirical P value from the random sampling.

**GO-term enrichment analysis**. The enrichment analysis was performed using the software Panther, version 13.0[70]. Differentially expressed genes were selected from Geeven et al.[29], fixing a threshold of P value < 0.05.

**Immunofluorescence**. Cells grown on glass coverslips (VWR) were fixed with 3.7% formaldehyde in PBS for 15 min and permeabilized with 0.5% Triton X-100 in PBS for 20 min. Samples were blocked in 3% BSA (Sigma Aldrich) in PBS before overnight incubation at 4 °C with primary antibodies: 1:400 S9.6 (gift from P. Huertas, CABIMER, Spain[47] or 1:200 anti-γH2AX (Abcam), followed by 1 h incubation at room temperature with the respective secondary antibodies conjugated to Alexa-Fluor 594, 647 or 488 (Thermo Fisher Scientific) and then 5 min staining at room temperature with 2 ng/μl of DAPI (Merck) in PBS. Coverslips were mounted in Prolong Diamond (Life Technologies) and visual acquisition was performed in a LSM510 AxioImager M1 microscope (Zeiss) using either a ×63 or a ×100 oil objective. Nuclear segmentation was based on DAPI staining. Statistical analyses were performed in Prism v5.0.4 (GraphPad Software) using the non-parametric Mann–Whitney rank sum test . *P < 0.05, **P < 0.01, ***P < 0.001, ****P < 0.0001.

**Nascent DNA and RNA labelling**. For nascent DNA labelling, cells were incubated in culture medium supplemented with 10 μM EdU for 20 min, rinsed in cold PBS before fixation in 3.7% formaldehyde for 15 min. EdU incorporation was revealed with Click-iT® EdU Imaging kits (Invitrogen) using Alexa Fluor 647 dye according to manufacturer's instructions. Coverslips were staining at room temperature with 2 ng/μl of DAPI (Merck) in PBS for 5 min and then mounted in Prolong Diamond (Life Technologies). Cells were analised using the ImageJ v1.51a software and scored as early-, middle-, or late-S phase according to their EdU replication-foci patterns. Statistical analyses were performed in Prism v5.0.4 (GraphPad Software) using the non-parametric Mann–Whitney rank sum test (Supplementary Figure 6). *P < 0.05, **P < 0.01, ***P < 0.001, ****P < 0.0001.

For nascent RNA labelling, cells were incubated in culture medium supplemented with 0.5 mM EU for 1 h and processed as above. EU incorporation was revealed with Click-iT® RNA Imaging kits (Invitrogen) using Alexa Fluor 488 dye according to manufacturer´s instructions. The mean EU fluorescence intensity per cell was obtained using ImageJ v1.51a software by averaging the mean grey value of EU signal measured in each nucleus. Nuclear segmentation was based on DAPI staining. Statistical analyses were performed in Prism v5.0.4 (GraphPad

Software) using the non-parametric Mann–Whitney rank sum test . *$P < 0.05$, **$P < 0.01$, ***$P < 0.001$, ****$P < 0.0001$.

**Pre-mRNA analysis.** Transient inhibition of RNAPII Ser2 phosphorylation with DRB and release was adapted from Singh and Padget (2009)[45] following the modifications of Jimeno-González et al.[46]. Briefly, subconfluent cell cultures were incubated in complete medium supplemented with 100 μM DRB for 3 h, then DRB-medium was washed-off and fresh medium was added to resume transcription elongation. Total RNA from equivalent number of cells was isolated every 10 min with RNeasy kit (Quiagen) following the manufacturer´s instructions. One microgram of RNA per time point was treated with DNAse (Roche), and reverse transcription reactions were performed with SuperScript III First Strand Synthesis System (Invitrogen) using random hexamers. Pre-mRNAs at the different time-points were quantified by RT-qPCR with primers spanning different exon–intron junctions (Supplementary Table 1). qPCR reactions were performed with an ABI Prism 7900HT Detection System (Applied Biosystems), with HotStar Taq polymerase (Qiagen) and SYBR (Molecular Probes). qPCR conditions were empirically tested for each primer pair and only those showing and slope of −3.3 +-10% and R2 value > 0.99 were accepted. All reactions were performed in duplicate in two independent experiments. Quantitative analyses were carried out using the ABI Prism 7900HT SDS Software (v 2.4). Pre-mRNA values were normalized to the values of the non-DRB-treated sample, which was set to one. Results are shown as means ± s.d.

**DNA/RNA immunoprecipitation.** DRIP experiments were performed with 0.4 μg/μl S9.6 antibody per IP reaction essentially as described in Skourti-Stathaki et al.[50]. Samples were treated either with 1 mg/ml RNAseA (Sigma), 30 U RNAseH (New England Biolabs), or both RNAseA and RNAseH, for 18 h at 37 ℃ before immunoprecipitation. DRIPs and input material were used as templates for qPCR reactions as described above. Primer sequences and PCR conditions are shown in Supplementary Table 1.

**Data availability.** The SNS-seq and replication timing datasets are deposited at the NCBI GEO (GSE99741). All relevant data are available from the authors.

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

## Acknowledgements

We are grateful to Arthur Skoultchi, Félix Prado, Juan Méndez, Pablo Huertas and Andrés Aguilera for reagents and advice, and to Fernando Azorín for sharing data with us before publication. We also thank Fernando Azorín, Juan Méndez and Félix Prado for comments on the manuscript; Silvia Jimeno-González and José Carlos Reyes for advice on transcription elongation experiments; Daniel Gómez and Alberto Ferrera for help in some stages of the project and members of the Crisanto Gutierrez and M.G. laboratories for discussions. This work was supported by the Spanish Ministry of Economy and Competitiveness, MINECO (BFU2013-45276-P and BFU2016-78849-P, co-financed by The European Union FEDER funds) to M.G., and by ANR grant (ANR-14-CE10-0008-02) and Ligue Contre le Cancer Comité de Paris (RS16/75-108) to J.-C.C., R.A. and J.M.F.-J. were supported by grants from the Portuguese Foundation for Science and Technology (SFRH/BD/81027/11) and MINECO FPI (BES-2014-070050), respectively.

## Author contributions

R.A., J.M.F.-J., C.S.-M., J.-C.C., L.C.-A., R.L. and G.H. performed experiments. J.M.F.-J. and J.-C.C. performed the computational analyses of SNS-Seq and replication timing datasets, respectively. R.A., J.M.F.-J., C.S.-M. and M.G. designed the experiments. A.A. contributed to HMGB1-KO MEF derivation, experimental design and analysis. M.G. conceived and analysed experiments and wrote the article. All authors analysed the data, discussed the results and approved the final version of the manuscript.

## Additional information

**Competing interests:** The authors declare no competing interests.

