## [Peer Review File · Nature Communications]

Reviewers' comments:

Reviewer #1 (Remarks to the Author):

The authors bring a variety of approaches including DNA fiber analysis and SNS sequencing for origin identification to characterise DNA replication patterns in cells in which chromatin 'compaction' has been altered by deleting specific genes. They show that deletion of HMGB1, which reduces nucleosome occupancy, has little effect on inter-origin distance, but leads to an increase in fork progression rate. Deletion of the three histone H1 genes, which causes a loss of chromatin compaction, severely affects initiation patterns and leads to replication fork collapse/stalling due to collisions with transcription complexes.

On the positive side, the experiments are well-designed, well-controlled and sensibly interpreted. On the negative side, the paper is largely descriptive, with little or no mechanistic insight. For example, is the disruption of initiation patterns in the triple H1 knockout a consequence of hyper-loading of MCM, more non-specific ORC binding, or some downstream step? Similarly, there is a correlation between the increase in fork 'stalling' and R loop formation, but a definitive experiment to show this would be to show suppression of the phenotype by overexpression of RNaseH, which has been done in a number of labs in other contexts. Nonetheless, I think this paper makes some interesting contributions to the replication field which will be of broad interest, and therefore publishable in Nature Comms. Because of the lack of mechanistic insight, however, I feel there are some places where experimental results are over-interpreted and need some qualification.

1. In general, the paper is written assuming the effects of chromatin changes are directly affecting DNA replication. However, the possibility that some or all the effects are caused by indirect changes to expression of genes – components of the replication machinery or regulators of replication – cannot be ruled out and need to be acknowledged and discussed.
2. It has been well-documented that there is an inverse relationship between number of origins firing and fork rate, and it has been postulated, with some evidence, that decrease in fork rate may increase origin firing via dormant origins, and conversely, reduction in origin firing may increase fork rate perhaps by affecting the dNTP pool. This needs to be discussed and it needs to be explicitly acknowledged that the experiments in this paper cannot distinguish between direct effects of chromatin on fork rate. In several places in the paper this would be testable – for example, is the reduced fork rate in H1-TKO cells suppressed by low concentrations of a DDK inhibitor?
3. The use of inter-origin distance as a measure of origin firing frequency is dangerous and probably incorrect. Inter-origin distance measures origin firing in local clusters but does not report on overall origin firing in cells.
4. The status of the DNA damage checkpoint is generally ignored in most experiments. Since this checkpoint affects origin firing, it needs to be at least discussed.
5. The authors cite references indicating that only a small number of genes are affected in the H1-TKO cells, yet they later show a significant increase in aberrant transcripts from genes. Is it possible that the original conclusion about minimal effects on transcription are flawed, and that, while overall transcription may be similar, the level of transcripts capable of encoding functional proteins may be significantly reduced? If so, this again raises the possibility of indirect effects via gene expression (point 1)

Minor point:

1. The references are neither in alphabetical order nor are they numbered which makes it very difficult to follow.
2. It might be interesting to compare the regions in the H1-TKO cells that show timing changes with the Gilbert data on domains that show changes in timing during ES cell differentiation.

Reviewer #2 (Remarks to the Author):

In this manuscript Almeida et al. give evidence on how chromatin compaction affects replication fork progression. This is a very interesting work that provides exciting insight into the relation between chromatin conformation and transcription/replication conflicts. Nevertheless, certain concerns need to be addressed in order for it to be suitable for publication.

Major points

- The authors need to validate the cell systems that they are using by providing WBs showing the HMGB1 protein levels in WT and HMGB1-KO MEFs, as well as H1 protein levels in WT and H1-TKO cells.
- In lines 116-119 the authors write: 'Note that the overlap between WT and HMGB1-KO ORIs (63%) is close to that obtained between replicates (78-80%) (Figure 1b), in spite of the differences in read coverage and consequently in ORI numbers between individual SNS-seq experiments (Supplementary Figure 1b).' It's not clear to me what the authors mean with this statement, and also 15-17% is not a close but a substantial difference. I would suggest that the authors rephrase this.
- The authors' main interest revolves around chromatin conformation/compaction. Therefore, they need to show upon HMGB1 KO and H1 reduction how is chromatin compaction affected. Histone H3 S10 phosphorylation (H3S10P) is a marker of chromatin condensation and may be used to show this.
- The authors make various speculative assumptions in order to interpret the results in Fig 2c-e and since they are interested in deciphering how reduced H1 causes replication stress they should address these hypotheses. Regarding the untimely origin firing as a possible cause of replication stress, the authors can inhibit origin firing by CDK inhibitor roscovitine or CDC7 inhibitor PHA-767491 and assess the effect on IOD, fork speed and fork asymmetry in WT and H1-TKO cells. With regards to depletion of dNTP pools, they can add ribonucleosides to the cells (ribonucleosides are precursors of deoxyribonucleosides and can be uptaken by the cells by adding them to the growth medium) and again assess the effect on IOD, fork speed and fork asymmetry in WT and H1-TKO cells.
- In lines 166-168 the authors write: 'To ensure that the decrease in fork rates was not skewed by the high percentage of stalled forks, we quantified separately the velocities of sister forks with less than 30% length difference (Supplementary Figure 2h).' This experimental is unnecessary as fork rate is the end result of anything that the replication machinery will encounter while traversing the DNA, stalled forks included. Not only that, but what is the scientific criterium to choose the sister forks with less than 30% length difference. Why 30% and not 20%?
- Regarding the use of the S9.6 antibody for the detection of R-loops. It has been shown that S9.6 antibody apart from R-loops has an affinity for dsRNA (PMID: 23784994). In order to exclude the possibility of non specific S9.6 staining the authors need to repeat the IF experiments of Figure 4d in the presence of RNase A that would not affect R-loops but degrade any dsRNA. Furthermore, since the authors raise a serious point on H1-TKO cells exhibiting increased R-loops they should further support their results with DRIP experiments (PMID: 21700224) that are more sensitive and accurate than S9.6 IF.
- The authors show that transcription inhibition rescues the replication stress in H1-TKO cells. Since, the authors have shown the induction of R-loops it would be interesting to assess the degree of involvement of R-loops in the H1-KO-induced replication stress. To do this the authors should overexpress an R-loop removing enzyme (such as RNaseH1 or Senataxin) in WT and H1-TKO cells and assess IOD, fork rate, fork asymmetry, S9.6 and γ H2AX intensity.
- In Figure 5a it's not clear what the incubation times of α -amanitin, ActD or α -amanitin + ActD are. Is it 6 h α -amanitin, 1 h ActD? And if so, α -amanitin + ActD is it consecutively or one after the other? The authors need to clarify this. Also, I don't understand what is the point of adding ActD in the first place, which is also known to induce DSBs (PMID: 15613478). In the figure legend, it says: 'to specifically evaluate the inhibition of RNAPII nascent transcription.' What does this mean? Expanding on this, in lines 249-250 the authors write: 'Cells were incubated with α -amanitin for 6 h, in conditions where RNAPII nascent transcripts were undetectable.'. I do not

understand what the authors mean with this phrase. The authors need to rephrase all of this.

- In Supplementary Figure 5f the authors provide quantification of EU fluorescence intensity of Figure 5a. They should provide these results into a plot and include it in Figure 5.
- In order to make safe comparisons regarding replication fork speeds and transcription elongation rates in the HCT-shSLBP1 cells +/- Dox, the authors need to measure the latter themselves and produce the results and not rely on publications of other groups. Also, the authors need to show the protein levels of SLBP, as well as H2AX, H2AZ and H3.3 on chromatin from HCT-shSLBP1 cells +/- Dox.

Minor points

- In Figure 4b the scale at Y axis in WT and H1-TKO at the Inpp5a gene should be the same at 4.
- In Figure 4c the authors should indicate that the representative images show EU staining.
- In Figure 5a the legend of the first image needs correction.

POINT-BY-POINT ANSWERS TO THE REVIEWERS:

REVIEWER#1.

1.- On the positive side, the experiments are well-designed, well-controlled and sensibly interpreted. On the negative side, the paper is largely descriptive, with little or no mechanistic insight. For example, is the disruption of initiation patterns in the triple H1 knockout a consequence of , more non-specific ORC binding, or some downstream step?

We agree with the reviewer's comment and, following her/his advice we have tested the levels of phosphorylated MCM2 (P-MCM2) in chromatin in mES H1-TKO cells and their WT counterparts by cell fractionation as a measure of replication origin activity (Figure 2e of the revised version). Unfortunately, antibodies against ORC proteins don't work reliably in mouse cells for unclear reasons. We found increased P-MCM2 levels in cells depleted of histone H1, confirming that more origins are activated in this chromatin context. Following-up from this, we conducted a series of experiments to unveil the causes of the replicative stress of this cells (see answer to point 4 and to Reviewer#2, point 4). These new results are described in pages 5-6, lines 141-176 and shown in Figure 2f-h of the revised version of the manuscript.

2-. Similarly, there is a correlation between the increase in fork 'stalling' and R loop formation, but a definitive experiment to show this would be to show suppression of the phenotype by overexpression of RNaseH, which has been done in a number of labs in other contexts.

We also acknowledged this criticism and performed transient transfection experiments with an RNaseH1-overexpression plasmid. We found that *in vivo* R-loop removal enhanced the rate of fork progression and partially recover fork asymmetry, supporting that R-loop accumulation promote the replication phenotype of H1-TKO cells. These new results are described in page 9, lines 284-292 and shown in Fig. 6e-i of the revised version of the manuscript.

3.- In general, the paper is written assuming the effects of chromatin changes are directly affecting DNA replication. However, the possibility that some or all the effects are caused by indirect changes to expression of genes – components of the replication machinery or regulators of replication – cannot be ruled out and need to be acknowledged and discussed.

To address this issue, we performed GO-term analyses of previously published RNA-Seq datasets derived from the same WT and H1-TKO mES (Geeven et al., 2015 Genome Biol: 16). No significant enrichment of cell cycle or DNA replication term classes was found within the differentially expressed genes. These results are shown in Suppl. Fig. 4c-d and described in page 6, lines 169-176.

4.- It has been well-documented that there is an inverse relationship between number of origins firing and fork rate, and it has been postulated, with some evidence, that decrease in fork rate may increase origin firing via dormant origins, and conversely, reduction in origin firing may increase fork rate perhaps by affecting the dNTP pool. This needs to be discussed and it needs to be explicitly acknowledged that the experiments in this paper cannot distinguish between direct effects of chromatin on fork rate. In several places in the paper this would be testable – for example, is the reduced fork rate in H1-TKO cells suppressed by low concentrations of a DDK inhibitor?

This is a relevant point also raised by the Reviewer#2 (point 3) that we didn't test in the previous version of our work. To address this in the revised version, we have added a new paragraph in the results section dedicated to decipher how reduced levels of histone H1 cause replicative stress (pages 5-6, lines 141-176). We found that increasing fork rates either by inhibiting origin firing with the CDC7 specific inhibitor PHA-767491 or by adding ribonucleosides improves fork asymmetry in H1-TKO cells (Figure 2f-h). These results were unanticipated and suggest that faster fork velocities can suppress the instability of ongoing forks in H1-depleted chromatin. We speculate that slow-moving forks in H1-TKO cells are highly sensitized to encounters with transcriptional roadblocks such as arrested RNAPII complexes or accumulated R-loops (lines 362, 367 of the Discussion section).

5.- The status of the DNA damage checkpoint is generally ignored in most experiments. Since this checkpoint affects origin firing, it needs to be at least discussed.

WB analysis of P-CBK1 and P-H2AX levels are now shown in Fig 2e of the revised version.

6.- The authors cite references indicating that only a small number of genes are affected in the H1-TKO cells, yet they later show a significant increase in aberrant transcripts from genes. Is it possible that the original conclusion about minimal effects on transcription are flawed, and that, while overall transcription may be similar, the level of transcripts capable of encoding functional proteins may be significantly reduced? If so, this again raises the possibility of indirect effects via gene expression (point 1)

Published work analysing the expression profile of protein-coding genes either by microarray hybridisation and, more recently, by RNA-seq identify a reduced number of genes showing differential expression in cells depleted of histone H1 (Fant et al., 2005 Cell:123; Murga et al., 2007 JCB: 278; Geeven et al., 2015 Genome Biol:16). Similar weak contribution of H1 on gene expression have been found by depleting H1 in Drosophila (Vujatovic et al., 2012 NAR:40) or the H1-like gene Hho in *Saccharomyces cerevisiae* (Hellauer et al., 2001 JBC:276). Given our results in altered RNAPII dynamics in H1-TKO cells, we propose that, indeed, the lack of H1 might enhance non-productive transcription initiation (page 8, lines 229-239), a possibility that we are currently testing in the laboratory.

Minor points:

1. The references are neither in alphabetical order nor are they numbered which makes it very difficult to follow.

Revised in numbered order as suggested.

2. It might be interesting to compare the regions in the H1-TKO cells that show timing changes with the Gilbert data on domains that show changes in timing during ES cell differentiation.

The comparison has been included in Suppl. Fig . 5b of the revised version.

REVIEWER#2

-1.- The authors need to validate the cell systems that they are using by providing WBs showing the HMGB1 protein levels in WT and HMGB1-KO MEFs, as well as H1 protein levels in WT and H1-TKO cells.

All throughout our work we used previously characterized cell lines, therefore we didn't consider necessary reproducing published results. HMGB1-KO cells have been characterized in Celona et al., 2011. PLoS Biol:9, and we derived MEFs from embryos of heterozygous crosses at the laboratory where the KO mice were initially generated; H1-TKO cells were obtained also from the lab which originally derived them and have been characterized from many laboratories along the years (Fan et al., 2003 MCB:23; Murga et al., 2007 JCB:278; Geeven et al., 2015 Genom Biol: 16) and HCT-shSLBP1 is a previously characterized stable human cell line obtained also from the laboratory who derived it (Jimeno-Gonzalez et al., 2015 PNAS:112). We referenced all these papers in the previous version of the manuscript. However, following the Reviewer's advice, we have included the following validations in the revised version of the manuscript:

- a. WB showing HMGB1 levels in WT MEFs and HMGB1-KO MEFs in Supplementary Figure 1a;
- b. IGV snapshots of SNS-seq read coverages in WT mES and H1-TKO mES showing the specific ablation of the H1c, H1d and H1e genes in Supplementary Figure 1f;
- c. WB showing SLBP levels in Dox-treated and control HCT-shSLBP.1 cells in Supplementary Figure 8a.

2.- In lines 116-119 the authors write: 'Note that the overlap between WT and HMGB1-KO ORIs (63%) is close to that obtained between replicates (78-80%) (Figure 1b), in spite of the differences in read coverage and consequently in ORI numbers between individual SNS-seq experiments (Supplementary Figure 1b).' It's not clear to me what the authors mean with this statement, and also 15-17% is not a close but a substantial difference. I would suggest that the authors rephrase this.

We agree this was unclear. The sentences have been re-phrased as suggested (page 4, lines 106-112 of the revised version).

3.- The authors' main interest revolves around chromatin conformation/compaction. Therefore, they need to show upon HMGB1 KO and H1 reduction how is chromatin compaction affected. Histone H3 S10 phosphorylation (H3S10P) is a marker of chromatin condensation and may be used to show this.

As argued in point 1, the chromatin compaction alterations in H1-TKO cells have been previously demonstrated by electron microscopy analysis in the original work characterizing this cell line (Fan et al., 2005 Cell:123). In the case of HMGB1-KO cells, we agree with the referee that neither previous authors nor we did directly address chromatin compaction and therefore, in the revised version of the manuscript, we carefully replace this term along the text. For example, in lines 122-124, we replaced the sentence "The data indicates that in chromatin with a more open conformation the replication initiation landscape is globally maintained and replication forks move faster and normally coordinated" for "The data indicates that in this context of more dynamic chromatin, the replication initiation landscape is globally maintained and replication forks move faster with stable elongation dynamics".

Following the Reviewer advice we evaluated chromatin condensation by IF with H3S10P antibodies. H3S10P clearly highlighted condensed chromosomes in mitotic cells without labelling interphase cells. See representative images for mES cells WT and H1-TKO below.

3.- The authors make various speculative assumptions in order to interpret the results in Fig 2c-e and since they are interested in deciphering how reduced H1 causes replication stress they should address these hypotheses. Regarding the untimely origin firing as a possible cause of replication stress, the authors can inhibit origin firing by CDK inhibitor roscovitine or CDC7 inhibitor PHA-767491 and assess the effect on IOD, fork speed and fork asymmetry in WT and H1-TKO cells. With regards to depletion of dNTP pools, they can add ribonucleosides to the cells (ribonucleosides are precursors of deoxyribonucleosides and can be uptaken by the cells by adding them to the growth medium) and again assess the effect on IOD, fork speed and fork asymmetry in WT and H1-TKO cells.

As detailed in the answer to Reviewer#1, point 4, we have performed the suggested experiments that are now shown in Fig. 2f-h and described in a new paragraph in pages 5-6, lines 156-176 of the revised manuscript.

4.- In lines 166-168 the authors write: 'To ensure that the decrease in fork rates was not skewed by the high percentage of stalled forks, we quantified separately the velocities of sister forks with less than 30% length difference (Supplementary Figure 2h).' This experimental is unnecessary

as fork rate is the end result of anything that the replication machinery will encounter while traversing the DNA, stalled forks included. Not only that, but what is the scientific criterium to choose the sister forks with less than 30% length difference. Why 30% and not 20%?

We have removed this experiment in the revised version as requested. We considered asymmetric sister forks those displaying more than 30% difference in fork velocities between the right and the left fork based on previous works (Tuduri et al., 2009 Nat Cell Biol:11; Fu et al., 2015 Nat. Commun:6).

5.- Regarding the use of the S9.6 antibody for the detection of R-loops. It has been shown that S9.6 antibody apart from R-loops has an affinity for dsRNA (PMID: 23784994). In order to exclude the possibility of non specific S9.6 staining the authors need to repeat the IF experiments of Figure 4d in the presence of RNase A that would not affect R-loops but degrade any dsRNA. Furthermore, since the authors raise a serious point on H1-TKO cells exhibiting increased R-loops they should further support their results with DRIP experiments (PMID: 21700224) that are more sensitive and accurate than S9.6 IF.

We concur with the Reviewer about the concerns on the specificity of the S9.6 antibody for the detection of DNA:RNA hybrids. Acknowledging her/his comment, we have performed IF experiments after treatment with RNase A and RNaseH (Fig. 4d of the revised version), as well as DRIP experiments after treatment of RNaseA, RNaseH, or both (Fig. 4e of the revised version). The results are described in a new paragraph in page 8, lines 232-260 of the revised manuscript. Both sets of experiments confirm the accumulation R-loops in H1-TKO cells.

6.- The authors show that transcription inhibition rescues the replication stress in H1-TKO cells. Since, the authors have shown the induction of R-loops it would be interesting to assess the degree of involvement of R-loops in the H1-KO-induced replication stress. To do this the authors should overexpress an R-loop removing enzyme (such as RNaseH1 or Senataxin) in WT and H1-TKO cells and assess IOD, fork rate, fork asymmetry, S9.6 and γ H2AX intensity.

See also answer to Reviewer#1 (point 2). We performed the requested experiments and found that R-loop removal upon RNaseH1-overexpression enhanced the rate of fork progression and partially recover fork asymmetry. These new results are described in page 9, lines 284-292 and shown in Fig. 6e-i of the revised version of the manuscript.

7.- In Figure 5a it's not clear what the incubation times of α -amanitin, ActD or α -amanitin + ActD are. Is it 6 h α -amanitin, 1 h ActD? And if so, α -amanitin + ActD is it consecutively or one after the other? The authors need to clarify this. Also, I don't understand what is the point of adding ActD in the first place, which is also known to induce DSBs (PMID: 15613478). In the figure legend, it says: 'to specifically evaluate the inhibition of RNAPII nascent transcription.' What does this mean? Expanding on this, in lines 249-250 the authors write: 'Cells were incubated with α -amanitin for 6 h, in conditions where RNAPII nascent transcripts were undetectable.'. I do not understand what the authors mean with this phrase. The authors need to rephrase all of this.

As requested, ActD experiments have been removed in the revised version of the manuscript. We included them in the previous version to inhibit RNAPI-dependent expression. The sentence "Cells were incubated with α -amanitin for 6 h, in conditions where RNAPII nascent transcripts were undetectable" has been rephrased as "Cells were incubated with α -amanitin for 6 h, conditions in which RNAPII nascent transcripts were undetectable" (lines 265-266).

8.- In Supplementary Figure 5f the authors provide quantification of EU fluorescence intensity of Figure 5a. They should provide these results into a plot and include it in Figure 5.

Done as suggested. Measurements of EU nuclear intensities upon α -amanitin treatment now are shown in Fig. 5a.

9.- In order to make safe comparisons regarding replication fork speeds and transcription elongation rates in the HCT-shSLBP1 cells +/- Dox, the authors need to measure the latter themselves and produce the results and not rely on publications of other groups. Also, the authors

need to show the protein levels of SLBP, as well as H2AX, H2AZ and H3.3 on chromatin from HCT-shSLBP1 cells +/- Dox.

See answer to point 1.

Minor points:

1.- In Figure 4b the scale at Y axis in WT and H1-TKO at the Inpp5a gene should be the same at 4.

The scale has been changed as suggested (Fig. 4b).

2.- In Figure 4c the authors should indicate that the representative images show EU staining.

Done as requested.

3.- In Figure 5a the legend of the first image needs correction.

Corrected as requested.

Reviewers' comments:

Reviewer #1 (Remarks to the Author):

The authors have addressed all of my concerns in this revised manuscript. I am happy to support publication.

Reviewer #2 (Remarks to the Author):

The authors have addressed my comments in my initial review and I find this revised version of the manuscript far superior to the previous one. Nevertheless, I have some concerns regarding the newly acquired data and their interpretation that need to be addressed.

-Regarding the H1-TKO-induced replication stress (decreased fork rate and increased fork asymmetry) the authors show a full rescue with RbNs supplementation, CDK inhibition, RNaseH1 overexpression and transcription inhibition. The question remains if the increased ORI activation (Fig 2a) is a cause (increased origins firing would lead to more collisions with transcription/Rloops) or a response to replication stress.

The authors use P-MCM2 as a marker of origin activity and test the effect of DRB, PHA and RbNs on it (Suppl. Fig 4a). In the H1-TKO samples, PHA as expected inhibits P-MCM2. Addition of RbNs also reduces P-MCM2 and since it also rescues the H1-TKO-induced replication stress (Fig 2f-h) it seems that collisions between non-productive transcription/Rloops is the main cause of the defects in replication fork rates and asymmetry and that the increased ORI activity is a response to this. Since DRB also rescues the H1-TKO replication stress (Fig 6c-d) and brings the IOD defect back to normal levels (Fig 6b), one would expect P-MCM2 to be inhibited/reduced. Strangely, that is not the case (Suppl Fig 4a). There seems to be a reduction of P-MCM2 but based on the MCM2 western blot and coomassie staining the DRB sample is underloaded in comparison to the untreated sample. Unless that DRB sample was collected at a later time point after DRB release. The authors need to comment on this in the text.

Upon interpreting all of the acquired data, I find that they point towards the idea that ORI activation is a response to the conflicts between replication and transcription/Rloops, rather than the cause. The authors clearly state this in the text:

Lines 175-177: Taken together, these analysis indicate that the strong alteration of the SNS landscape detected in H1-TKO cells reflects the firing of extra ORIs as well as a widespread accumulation of stalled replication intermediates.

Lines: 360-363: The opposite situation is exemplified by the reduction of histone H1 levels in chromatin, which triggers a dramatic alteration in replication initiation patterns, including the firing of additional ORIs and massive fork stalling as results from elevated transcription-replication conflicts (Figs. 2, 5 and 6).

However, in the abstract they claim that increased ORI activity is a cause and not a result of increased replication-transcription collisions.

Lines 39-42: We find that loss of chromatin compaction in H1-depleted cells massively disrupts the replication initiation patterns, triggering the accumulation of stalled forks and DNA damage as a consequence of transcription-replication conflicts.

The authors need to clarify this, as it is an important part of their story.

-The authors have performed the experiments of RNaseH1 overexpression that I asked for in my initial review, which they included in Fig 6e-i. However they need to show in a way the overexpression of RNaseH1 (eg. western blot).

-In Fig 2e the plot needs to show what the blue and red colours stand for.

-There is a spelling mistake at the end of line 359. The word 'does' should be replaced by 'do'.

Response to Reviewers' comments:

We thank both reviewers for their detailed and critical comments and are very glad that they found our work improved after the revision. We have now further revised our manuscript in response to the additional points raised by Reviewer#2.

1. Regarding the H1-TKO-induced replication stress (decreased fork rate and increased fork asymmetry) the authors show a full rescue with RbNs supplementation, CDK inhibition, RNaseH1 overexpression and transcription inhibition. The question remains if the increased ORI activation (Fig 2a) is a cause (increased origins firing would lead to more collisions with transcription/Rloops) or a response to replication stress.

We agree that discerning whether increased ORI activation in H1-TKO cells is a cause or a consequence of the elevated replication-transcription conflicts is an interesting question. We didn't address it directly in the previous version of the manuscript as it is unclear how much of the altered replication initiation pattern detected in the SNS-experiments (Fig 2a) is due to the firing of extra origins, or both to the firing of extra origins and to the accumulation of short stalled replication intermediates. For this reason, as mentioned by the reviewer in her/his comment below, we stated in the text (lines 175-177) that "these analysis indicate that the strong alteration of the SNS landscape detected in H1-TKO cells reflects the firing of extra ORIs as well as a widespread accumulation of stalled replication intermediates". Following the reviewer's suggestion, we have now added a paragraph in the Results (lines 283-290) and in the Discussion sections (lines 394-398) addressing this issue. See also the reply to the following points.

2. The authors use P-MCM2 as a marker of origin activity and test the effect of DRB, PHA and RbNs on it (Suppl. Fig 4a). In the H1-TKO samples, PHA as expected inhibits P-MCM2. Addition of RbNs also reduces P-MCM2 and since it also rescues the H1-TKO-induced replication stress (Fig 2f-h) it seems that collisions between non-productive transcription/Rloops is the main cause of the defects in replication fork rates and asymmetry and that the increased ORI activity is a response to this. Since DRB also rescues the H1-TKO replication stress (Fig 6c-d) and brings the IOD defect back to normal levels (Fig 6b), one would expect P-MCM2 to be inhibited/reduced. Strangely, that is not the case (Suppl Fig 4a). There seems to be a reduction of P-MCM2 but based on the MCM2 western blot and

coomassie staining the DRB sample is underloaded in comparison to the untreated sample. Unless that DRB sample was collected at a later time point after DRB release. The authors need to comment on this in the text.

The protein extracts used for the WB analysis shown in Suppl. Fig. 4a were taken in parallel to the preparation of the DNA fibers shown in Fig 2f-h and are representative of two independent experiments. We don't have a satisfactory explanation for the subtler reduction in P-MCM2 levels in H1-TKO cells upon DRB treatment in comparison to the reductions obtained upon PHA or RbNs treatments. We have indicated this explicitly in lines 283-286 of the revised version. It is important to note, however, that PHA and RbNs treatments also results in increased IODs and fork rates in WT cells without major reductions in P-MCM2 levels as charged by WB, suggesting that decreased origin activity can occur with minor variations in P-MCM2 levels.

Indeed, given our findings that; (i) transcription inhibition by DRB rescues all the replicative defects of H1-TKO cells without altering fork dynamics in WT cells and, (ii) transcription inhibition by α -amanitin and R-loop degradation by RNH1 overexpression rescue both fork rates and fork asymmetry in H1-TKO cells without increasing IODs, we favour the hypothesis, in line with the reviewers', that conflicts between replication and transcription/R-loops are the main cause of the replicative stress of cells with reduced amounts of histone H1, and that increased origin activity is a response to this. Quite likely, the increased ORI activity arising in this condition might also enhance the replicative stress state by further increasing the probability of collisions or by favouring R-loop formation. We have included a paragraph commenting this possibility in page 9, lines 286-290 of the Results section and in the Discussion (page 12, lines 394-398).

3. Upon interpreting all of the acquired data, I find that they point towards the idea that ORI activation is a response to the conflicts between replication and transcription/Rloops, rather than the cause. The authors clearly state this in the text:

Lines 175-177: Taken together, these analysis indicate that the strong alteration of the SNS landscape detected in H1-TKO cells reflects the firing of extra ORIs as well as a widespread accumulation of stalled replication intermediates.

Lines: 360-363: The opposite situation is exemplified by the reduction of histone H1 levels in chromatin, which triggers a dramatic alteration in replication initiation patterns, including the firing of additional ORIs and massive fork stalling as results from elevated transcription-replication conflicts (Figs. 2, 5 and 6). However, in the abstract they claim that increased ORI activity is a cause and not a result of increased replication-transcription collisions.

Lines 39-42: We find that loss of chromatin compaction in H1-depleted cells massively disrupts the replication initiation patterns, triggering the accumulation of stalled forks and DNA damage as a consequence of transcription-replication conflicts. The authors need to clarify this, as it is an important part of their story.

We agree that the sentence in lines 39-42 of the abstract could be misleading and we have re-written it accordingly. In addition, as mentioned above, we've now added a dedicated paragraph in the Results and Discussion sections speculating about the possible interpretation of all the data.

4. The authors have performed the experiments of RNaseH1 overexpression that I asked for in my initial review, which they included in Fig 6e-i. However they need to show in a way the overexpression of RNaseH1 (eg. western blot).

We have now included the WB analysis of RNase H1 levels in cells transfected with empty vector or RNH1 overexpressing plasmid as requested (Fig 6e).

5. In Fig 2e the plot needs to show what the blue and red colours stand for.

The legend of the plots has been added as requested.

6. There is a spelling mistake at the end of line 359. The word 'does' should be replaced by 'do'.

Corrected as suggested.

REVIEWERS' COMMENTS:

Reviewer #2 (Remarks to the Author):

My concerns have been met and I find the manuscript suitable for publication in Nature Communications.

We are very happy that both reviewers found our work suitable to be published in Nature Communications. Please find their final comments below.

REVIEWERS' COMMENTS:

Reviewer #1 (Remarks to the Author):

The authors have addressed all of my concerns in this revised manuscript. I am happy to support publication.

Reviewer #2 (Remarks to the Author):

My concerns have been met and I find the manuscript suitable for publication in Nature Communications.